# Deep Natural Language Feature Learning for Interpretable Prediction

**Felipe Urrutia[1,2], Cristian Buc[1], Valentin Barriere[1,2]**
[1]Centro Nacional de Inteligencia Artificial, Macul, Chile
[2]Department of Computer Science, Universidad de Chile, Santiago, Chile
`furrutia@dim.uchile.cl, name.lastname@cenia.cl`

## Abstract

We propose a general method to break down a main complex task into a set of intermediary easier sub-tasks, which are formulated in natural language as binary questions related to the final target task. Our method allows for representing each example by a vector consisting of the answers to these questions. We call this representation Natural Language Learned Features (NLLF). NLLF is generated by a small transformer language model (e.g., BERT) that has been trained in a Natural Language Inference (NLI) fashion, using weak labels automatically obtained from a Large Language Model (LLM). We show that the LLM normally struggles for the main task using in-context learning, but can handle these easiest subtasks and produce useful weak labels to train a BERT. The NLI-like training of the BERT allows for tackling zero-shot inference with any binary question, and not necessarily the ones seen during the training. We show that this NLLF vector not only helps to reach better performances by enhancing any classifier, but that it can be used as input of an easy-to-interpret machine learning model like a decision tree. This decision tree is interpretable but also reaches high performances, surpassing those of a pre-trained transformer in some cases. We have successfully applied this method to two completely different tasks: detecting incoherence in students' answers to open-ended mathematics exam questions, and screening abstracts for a systematic literature review of scientific papers on climate change and agroecology.[1]

## 1 Introduction and Related Work

The use of AI models is becoming increasingly pervasive in today's society. As such, applications of these models have seen the light in domains where decisions can have dramatic consequences such as healthcare (Norgeot et al., 2019), justice (Dass et al., 2022), or finances (Heaton et al., 2017). This pervasiveness is intrinsically linked to the huge success of Deep Learning models, which have shown to scale particularly well to complex decision-making problems. However, this success comes at a cost. To solve complex (high-stakes) decisions, these models develop inner hidden representations which are hard to understand and interpret even by researchers implementing the models (Castelvecchi, 2016).

Regulations like the European Union's "Right to Explanation" (Goodman and Flaxman, 2017) or Russell et al. (2015) requirements for safe and secure AI in safety-critical tasks fostered advances in explainability. Therefore, there has been a growing interest in developing techniques allowing to explain the mechanisms subtending these so-called "black-box" models (Guidotti et al., 2018; Fel et al., 2022). Notably, this endeavor has given rise to an entire research field called Explainable AI (XAI), which focuses on providing human-interpretable information on the models' behavior (Gunning et al., 2019; Arrieta et al., 2020).

Explainable Deep Learning in Natural Language Processing (NLP) can be decomposed in two categories: representational and practical. Representational XAI in NLP focuses on understanding the underlying structure of the representations. For instance, studies have shown that Transformer-based architectures develop abstract symbolic, or compositional, representations (Lovering and Pavlick, 2022; Li et al., 2022b). Similarly, it has been shown that conceptual knowledge can have sparse representations, which can thus be located and edited to induce different predictions (Meng et al., 2022). Practical methods can analyze the outputs of the models, for example when perturbing the inputs (Tulio Ribeiro et al., 2016; Fel et al., 2023; Lundberg and Lee, 2017). Recently, practical XAI in NLP focuses on prompting to increase explainability. For instance, chain-of-thought prompting is a

---

[1]Code available at `https://github.com/furrutiav/nllf-emnlp-2023`

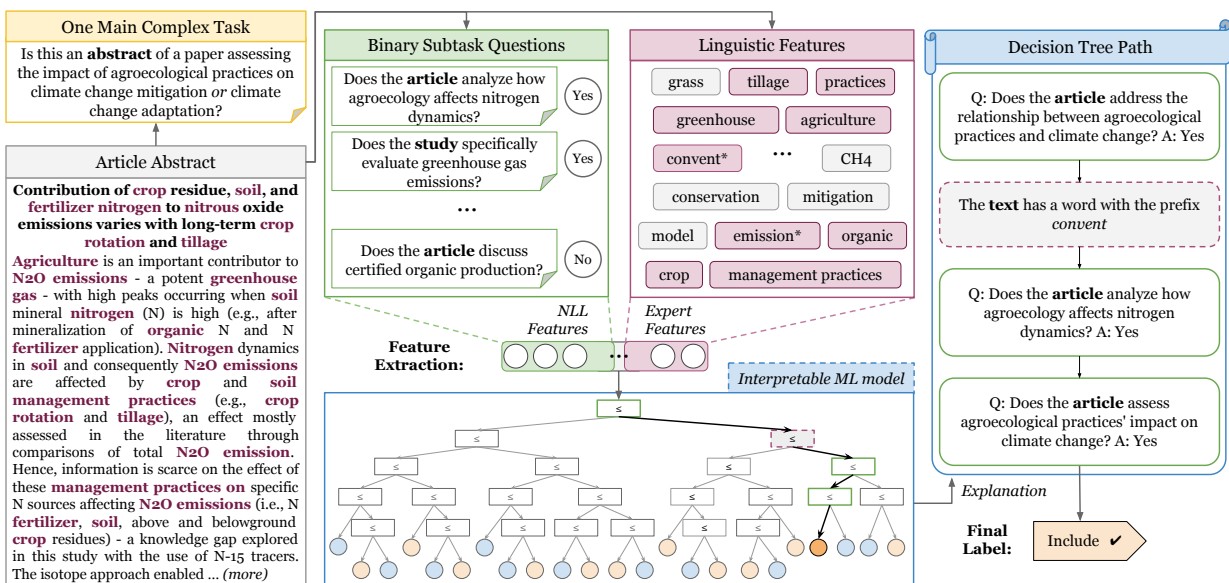

Figure 1: Overview of the proposed system: Extraction of Natural Language Learned Features and Expert Features in order to understand the decision process of an interpretable model for complex task solving.

method that implements a sequence of interposed NLP steps leading to a final answer (Wei et al., 2022; Wang et al., 2022; Zhao et al., 2023; Lyu et al., 2023). This method has the advantage to provide insights on the logical reasoning steps behind a model's behavior, and thus allows to understand (at a higher level) the predictive success or failure of LLM (Zhou et al., 2022; Diao et al., 2023; Wang et al., 2022).

Although advances in XAI within the NLP domain have offered interesting insights on the underlying mechanisms and representational structures of LLM, there has been a recent push to solely focus on interpretable models for high-stakes decisions (Rudin, 2019). This push is motivated by dramatic errors made by these models in real life situations, such as assessing criminal risk at large scale (Angwin et al., 2016) or incorrectly denying bail for criminals (Wexler, 2017). The main rationale behind this idea holds in that there will always be a certain level of error (or information loss) associated to the explanations of black-box models. Indeed, these explanations can, by definition, only partially incorporate information of the model's reasoning process.

Following Rudin (2019), we make the difference between an explainable black-box model and an interpretable white-box model. Explainability relies on algorithms aiming to explain the model predictions by showing cues to the user like LIME, or other ad-hoc methods (Fel et al., 2022; Colin et al.,

2022). Interpretability relies on the possibility to know exactly why the model is making a prediction because they are inherent to the prediction and faithful to what the model actually computes. However, methods like CoT which should be interpretable (because outputting explanations with their predictions) have not always shown to give faithful explanations (Radhakrishnan et al., 2023). It is also arguable that our method relies on learned representations from a BERT, which decreases its overall interpretability.

**Motivation and Contributions** In this work, we aim to reconcile the abilities of black-box LLM and interpretable machine learning (ML) models, by leveraging the impressive zero-shot abilities of LLM, instructed (Peng et al., 2023; Ouyang et al., 2022; Chung et al., 2022) or not (Wei et al., 2021), to improve the performance of ML models in more complex tasks. One particularly stunning feature of LLM is compositional systematicity (Lake and Baroni, 2018; Bubeck et al., 2023): the ability to decompose concepts into their constituent parts that can be recombined to produce entirely novel concepts. Such compositional representations is at the basis of systematic generalization in novel contexts (Brown et al., 2020). For instance, experimental work has shown that LLM are zero-shot learners, and can further improve this skill when encouraged to reason sequentially (Kojima et al., 2022a).

In our approach schematize in colors on Figure 1, we leverage the ability of LLM to decompose

complex tasks in simpler sub-tasks, and use these sub-tasks with a medium-size language model to create interpretable features (the *Binary Subtask Questions* (BSQ) and *Natural Language Learned Features* (NLLF) in green) and to improve the performance of ML classifiers. This classifier can be a simple interpretable model such as a Decision Tree with a readable decision path (in blue). The uniqueness of our work is to combine the strength of LLM and the explainability of ML classifiers, as opposed to similar previous work which as mainly focused on leveraging LLM to increase the reasoning abilities of smaller LM (Li et al., 2022a).

This work makes three main contributions. First, compared with chain-of-thoughts methods, which are often computationally expensive and are effective in solving certain types of reasoning problems, our approach is a computationally cheap and universal solution. Indeed, it can be applied to any problem that can be reasonably decomposed in simpler tasks (see methods below). Second, we present a method that allows simpler and interpretable models to solve complex reasoning tasks; tasks which are usually outside the realm of solutions for these type of models. Third, we show that those interpretable models can surprisingly outperform state-of-the-art LLM models in reasoning-based classification tasks.

To demonstrate the usefulness of the proposed method we focus on two different languages tasks that requires high levels of reasoning, and can be decomposed in subtasks with lower difficulty levels of reasoning, in English and in Spanish. We aim to show the generalization power of our method by *(i)* classifying the coherence of fourth grade students' answers to mathematical questions (Urrutia Vargas and Araya, 2023; Urrutia and Araya, 2023) (IAD: *Incoherent Answer Detection*), *(ii)* classifying scientific papers regarding a topic of interest in the context of a systematic literature review about agroecology and climate change (SAC: *Scientific Abstract Classification*).

## 2 Methods

This section describes the different parts of the proposed method, which are summarized in Figures 2 and 3. Subsection 2.1 shows how to utilize an instructed LLM to generate natural language binary subtasks questions that can be useful to solve a more complex task. Subsection 2.2 (in green in Figure 3) explains the process to leverage the

zero-shot ability of a LLM in order to label examples regarding the binary subtasks. Next, subsection 2.3 (in orange in Figure 3) contains details on how we train a BERT-like model (Devlin et al., 2018; Cañete et al., 2020) in a natural language inference (NLI) fashion to resolved those subtasks. Finally, subsections 2.4 and 2.5 (in blue in Figure 3) describe the process to generate interpretable representations and how to integrate them in an explainable model to solve the main task.

### 2.1 Lower-level Subtasks Generation

In this step, we are generating $C$ Binary Subtask Questions (BSQs), which are using a LLM. This step is not mandatory as a human practitioner could do it manually.

In order to identify subtasks of the main task, we randomly select a small percentage $p_q$ of samples from the training set. With the help of a LLM that we prompt using an instruction-based template (visible in Appendix E), we generate a set of 5 basic binary questions per sample useful to solve the main task as shown in Figure 2.

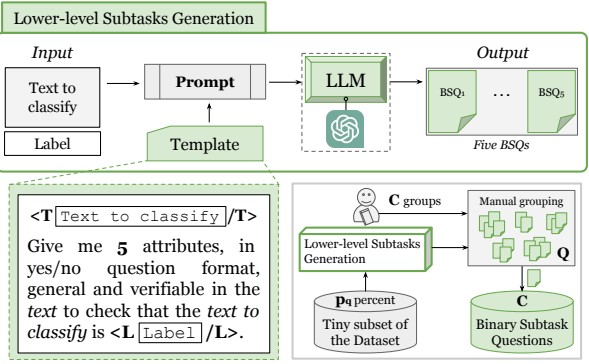

Figure 2: Automatic generation of BSQs using prompt templates with an LLM and manual grouping.

This process lead to a large set of $Q$ binary questions obtained from the $p_q$ subpart of the dataset. In order to reduce the redundancy in this large set, similar questions were manually grouped together into $C$ groups, and each group was reformulated into a unique general question and verifiable yes/no binary questions. This process leaves us with a set of $C$ questions.

### 2.2 Zero-shot Subtasks Labeling with an LLM

In this phase, we generate labels on some of the training examples regarding the $C$ subtasks. In order to achieve this, we are leveraging the zero-shot

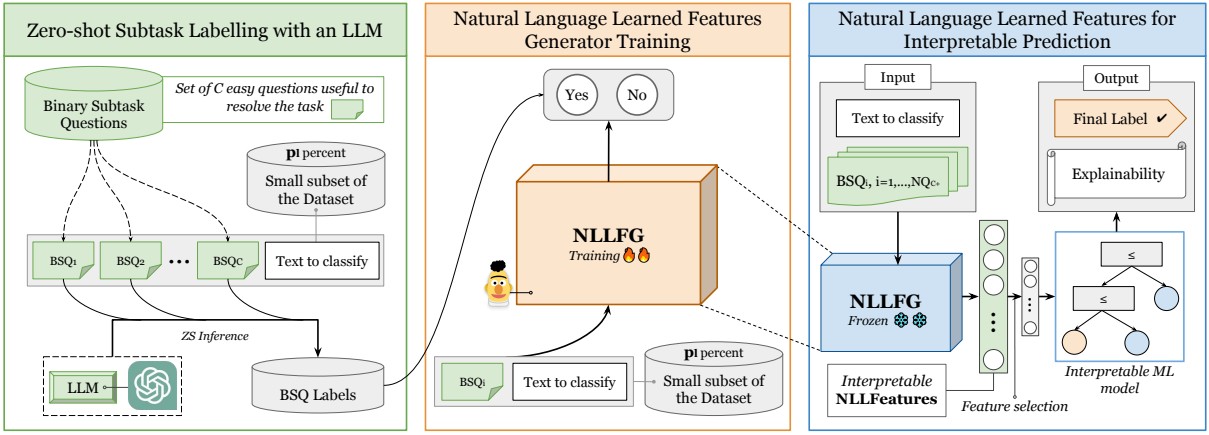

Figure 3: Full process of subtask labelisation, NLLFG training, NLLF generation and integration.

learning capacity of LLM to solve simple tasks. By prompting an LLM with samples from the training dataset with a low-level subtask binary question, we are able to annotate each example according to the $C$ binary subtasks. In the end, we obtain synthetics labels on a limited percentage $p_l$ of the training set, which totals to $C \times p_l$ percents of the initial training dataset size. The template of the prompts can be seen in Appendix E.

### 2.3 Natural Language Learned Feature Generator Training

In this stage, we use the examples tagged with low-level weak labels obtained through a large model, to fine-tune a smaller BERT-like transformer model[2] in a natural language inference (NLI) way. Given that the BSQs can be expressed and inserted in natural language inside the transformer, an NLI-type of inference means that the text to classify and the BSQ are seen as premise and hypothesis. It has two strong advantages: *(i)* it leverages the semantic knowledge encoded during pre-training to understand the label, *(ii)* it can be applied in a zero-shot manner using new labels formulated as natural language binary question (Yin et al., 2019; Vamvas and Sennrich, 2020; Barriere and Jacquet, 2022).

In the end, this model is able to predict, for every pair of sample associated with any binary question, if the answer to the question is *yes* or *no*. We call this model Natural Language Learned Feature Generator (NLLFG). More details are available in Appendix A.

[2]which has 1000 times less parameters than the LLM used beforehand

### 2.4 Natural Language Learned Feature Generation

**BSQ augmentation**   Only $C$ question were used for the training of the NLLFG because of budget cost, but way more might be used to represent a sample as the NLLFG can generate an answer to a question never seen during training, in a zero-shot way. Hence, we augment the set of questions by adding new questions from the expert domain: we used all the available BSQs before the manual clustering, translation of expert linguistics features into natural language, paraphrases of the $C$ BSQs and human-made BSQs. This leads to a set of $C_+$ questions.

**NLLF construction**   For every example, we use the NLLFG with the $C_+$ binary questions in order to create a vector of NLLF. The vector of NLLF was constructed by taking the sigmoid of the logit instead of the softmax, in order to keep the information about the confidence of the prediction for both classes (i.e., sometimes the predictions are far away from the decision hyperplan for both classes). Which means that for each BSQ, there are two values between 0 and 1: one representing the probability of the *Yes* answer, and one representing the probability of the *No* answer. This gave us a NLLF of size $2C_+$.

**Feature selection**   Finally, we ensure to select only the most effective features by removing the ones predicting non useful representations with feature selection. We employed a genetic algorithm for feature selection (Fortin et al., 2012), using decision trees as backbones. We executed the algorithm in a 15-fold cross-validation setting and selected the features that were selected at least one third of

the times. In other words, we removed the BSQs that lead to unsure answers from the model.

## 2.5 NLLF-boosted Explainable Model

the NLLF vector generated from a sample is a controlled representation, which can then be added to augment any existing classifier. In this work, given that we are focusing on interpretable modelization, we chose to use it as input of a Decision Tree (Breiman et al., 1984). We also enhanced the NLLF representation with Expert Features (EF) derived from linguistic patterns, which are known to be very precise but generalize poorly. In this way, we take the best of both world with an hybrid model benefiting from the robustness and high accuracy of deep transformers as well as the fine-grained precision of linguistic rules (Barriere, 2017).

## 3 Experiment and Results

### 3.1 Datasets

We used two datasets in order to validate our model. First, we present a dataset of abstracts and titles of English scientific articles, labeled regarding their pertinence towards a systematic literature review on Climate Change and Agroecology. Second, we present a dataset of coherent and incoherent students answers to open-ended questions of a mathematical test.

**Scientific Abstract Classification**  To evaluate our method on complex text, we use a dataset annotated in the context of a systematic literature review about the impact of agroecological practices on climate change mitigation and climate change adaptation.[3] More than 15k articles were retrieved from the Web of Science database using an extensive set of keywords related to Agroecology and Climate Change. The first 2,000 articles were tagged by two annotators, using the title and abstract of the article, regarding whether or not the article was relevant for the systematic literature review. If there was no consensus between the two annotators, a third annotator was called to arbitrate, which happened the case 14% of the time. The articles with missing abstracts were removed from this study, which left a total of 1,983 articles, from which 50.1% labeled as included and 49.9 % labeled as excluded.

---

[3]article to be published from the Agroecology research group of Sant'Anna di Pisa: `https://www.santannapisa.it/en/centro-di-ricerca/scienze-delle-piante/agroecology`

**Incoherent Answer Detection**  To evaluate our method on special domain text, we focus on the task of coherence detection in students' answers to an open-ended mathematical test questions. We used the dataset of Urrutia Vargas and Araya (2023) composed of 15,435 answers to 700+ different open-ended questions collected using the online e-learning platform ConectaIdeas. The answers' (in)coherence were manually annotated by several teachers. The test set only contained examples that were annotated similarly by at least three annotators. Both the train and test datasets are imbalanced between the classes, with respectively 13.3% and 20.1% of incoherent examples.

### 3.2 Baselines and proposed methods

In this section, we describe the baseline models that we evaluated in our experiments.

**Vanilla ChatGPT**  Because this model is known to have good performances at zero- and few-shot inference, we evaluate it in a 0/4-shot prompt strategies.

**CoT ChatGPT**  Chain-of-Thought has been shown to improve the performances of LLM for reasoning tasks. Hence, we enhance the model with this technique. We used the technique of Kojima et al. (2022b) for the zero-shot CoT.

**Self-ask ChatGPT**  Self-ask (Press et al., 2023) enhances compositional reasoning by explicitly formulating and answering follow-up questions before addressing the initial query to significantly reduces the compositionality gap.

**BERT**  We evaluate different models based on a BERT transformers (Devlin et al., 2018), processing the raw text. The Vanilla version connects one fully-connected layer after the [CLS] token. The other versions concatenate (previously extracted) expert features and NLLF with the [CLS] representation. For the IAD dataset, which is in Spanish, we used the Spanish version of BERT called BETO (Cañete et al., 2020).

**Decision Tree**  Decision trees were used as explainable models, with low height and only interpretable features. We used the same features as for BETO, plus added Bag-of-N-Grams (BoNG, variant of the Bag-of-Words; Harris, 1954) in order to model the text content.

| Model | Variant | Params | Explainability | IAD | | | SAC | | |
|---|---|---|---|---|---|---|---|---|---|
| | | | | Prec. | Rec. | F1 | Prec. | Rec. | F1 |
| ChatGPT | 0-shot | $\sim 10^{11}$ | ✗ | 19.70 | 76.47 | 31.33 | 76.57 | 50.27 | 35.23 |
| | 4-shots | | ✗ | 24.80 | 90.44 | 38.92 | 66.92 | 51.66 | 38.92 |
| | 0-shots CoT | | ✓ | 23.14 | 84.56 | 36.33 | 44.93 | 46.31 | 41.59 |
| | 4-shots CoT | | ✓ | 42.18 | 85.29 | 56.45 | 65.00 | 63.26 | **62.72** |
| | 0-shots SA | | ✓ | 21.29 | 82.35 | 33.84 | 63.65 | 55.15 | 48.13 |
| | 4-shots SA | | ✓ | 51.71 | 77.94 | **62.17** | 70.31 | 62.42 | 59.50 |
| BERT | Vanilla | $\sim 10^{8}$ | ✗ | 58.47 | 78.68 | 67.08 | 67.74 | 67.80 | 67.72 |
| | EF | | ✗ | 78.40 | 72.06 | 75.10 | 67.65 | 66.93 | 66.90 |
| | NLLF | | ✗ | 67.10 | 76.47 | 71.48 | 68.97 | 68.98 | 68.75 |
| | NLLF+EF | | ✗ | 80.49 | 72.79 | **76.45** | 73.66 | 73.61 | **73.63** |
| Decision Tree | BoNG | $\sim 10^{2}$ | ✓ | 100.0 | 8.09 | 14.97 | 65.38 | 65.13 | 65.15 |
| | EF | | ✓ | 83.33 | 66.18 | 73.77 | 68.18 | 66.49 | 64.95 |
| | NLLF | | ✓ | 75.00 | 44.12 | 55.56 | 62.41 | 62.43 | 62.25 |
| | NLLF+EF | | ✓ | 85.22 | 72.06 | **78.09** | 68.02 | 68.01 | **67.75** |
| | NLLF+BoNG | | ✓ | 82.28 | 47.79 | 60.47 | 66.21 | 66.26 | 66.20 |
| | NLLF+EF+BoNG | | ✓ | 85.22 | 72.06 | **78.09** | 68.17 | 67.43 | 67.41 |

Table 1: Precision, Recall and F1-score of all the configurations and models for Incoherent Answer Detection (IAD); and (Macro) Precision, Recall and F1-score of all the configurations and models for the Scientific Abstract Classification (SAC). Using Expert Features (EF), NLLF, and Bag-of-N-Grams (BoNG).

## 3.3 Experimental Protocol

### 3.3.1 Dataset splitting

**Scientific Abstract Classification** We randomly split the data into a training, a validation and a test sets following the proportion 70/10/20.

**Incoherent Answer Detection** We train the classifiers on the 2019 data and tested on a sample of 677 perfect-labeled answers from the 2017 dataset. The study used the different open-ended questions and answers, but the same definition of incoherence throughout, despite different students and teachers in each year.

**NLLFG training** For each task, we randomly split 90% of the weakly labeled examples into a training set and keep the last 10% for the model validation.

### 3.3.2 Evaluation Metrics

**Scientific Abstract Classification** As both the classes are important, we have adopted the macro-averaged precision, recall, and F1-score metrics as our evaluation criteria.

**Incoherent Answer Detection** To maintain consistency with the aforementioned work (Urrutia Vargas and Araya, 2023; Urrutia and Araya, 2023), we have adopted precision, recall, and F1-score metrics for the positive class (incoherent) as our evaluation criteria.

### 3.3.3 Method parameters

**Scientific Abstract Classification** We used a $p_q$ of 1.3% (21 examples) to generate BSQ and a $p_l$ of 10% to train the NLLFG. $C$ was set up to 13 questions, and the number of questions for the generation $C_+$ was 109 questions.

**Incoherent Answer Detection** We used a $p_q$ of .15% (21 examples) to generate BSQ and a $p_l$ of 10% to train the NLLFG. $C$ was set up to 10 questions, and the number of questions for the generation $C_+$ obtained was 66.

### 3.3.4 Implementation

The `transformers` library (Wolf et al., 2019) was used to access the pre-trained model and to train our models. We used BERT and BETO[4] as backbones for the NLLFG. The decision trees were trained using scikit-learn (Pedregosa et al., 2012). We used the 03/23/23 version of ChatGPT (Ouyang et al., 2022) as LLM. Other details can be found in Appendix B and C.

### 3.4 Results

The results are visible in Table 1 for respectively the IAD task and the SAC task. The last six columns detail the precision, recall and F1-score of the "*incoherent*" class for the IAD task, and the macro precision, recall and F1-score for the SAC task. In both cases, the best results overall are

---

[4]`bert-base-cased` and `bert-base-spanish-wwm-cased`

the ones from models enhanced by NLLF and EF, reach the F1-scores of 78.09% and 73.63%.

**ChatGPT**   For the SAC task, ChatGPT models display high precision, but overall low recall, and very low F1-score coming from a low F1-score for the exclude task, except for the 4-shot + CoT /SAC versions that reach a F1-score of 62.72% / 59.50%. This is because ChatGPT tends to categorize almost all of the articles as included. For the IAD task, ChatGPT models display poor precision and F1-score metrics, but overall high recall, meaning these models tend to categorize most of the answers as incoherent. Moreover, as expected, the 4-shots + SAC variant outperform all other Chat-GPT variants in F1-score (62.17%).

**BERT**   BERT-like models display the high metrics across the board in precision, recall and F1-score.  In particular, the models incorporating both NLLF and EF obtains the highest overall F1-score (76.45% and 73.63%) in both tasks. Surprisingly, enhancing the transformer with NLLF (resp. EF) provokes a drop in the performances for the IAD (resp. SAC) task. Note however, that BERT models belong to the class of models that are not explainable.

**Decision Tree**   The DT models also reached high performances across the board (with the exception of the BoNG variant for the IAD task). Specifically, the variant using NLLF+EF displays highest F1-score (78.09% and 67.75%) in that model class, and it is notable that adding BoNG features does not improve the performances. The DT models are simple and fully interpretable, and significantly outperforms a LLM like ChatGPT, while reaching performance metrics competitive with a deep learning (black-box) model. This approach provides interpretable steps to explain decision making within the tree (see Appendix H). The DTs using EF have very competitive results. Even though it is not relying on deep neural nets, it needs many complex handcrafted features coming from expert knowledge (Table 9).

## 4   Model Analysis

### 4.1   NLLF Accuracies

We quantify the error of the output of the decision tree using classical metrics, but not the error on the input of the tree, which is the error when creating the NLLF. Here we analyze how accurate were the

NLLF generated by the BERT-like model, and also the weak labels by the LLM.

**NLLFG Training**   We analyze the performance of the NLLFG on the validation set during it's training in order to quantify how good a NLI-like BERT transformer can reproduce the weak labels of an LLM. From the results shown in Table 2, we can see that the performances for the IAD task are much higher than the ones of the SAC task.

| Task | Label | Prec. | Rec. | F1 | Acc. |
|------|-------|-------|------|----|------|
| SAC  | Yes   | 74    | 86   | 79 | 73   |
|      | No    | 71    | 52   | 60 |      |
| IAD  | Yes   | 97    | 96   | 97 | 95   |
|      | No    | 92    | 94   | 93 |      |

Table 2: NLLFG performance on the validation set during the weak label training.

Nevertheless, we can see that the NLLFG tend to overclassify the examples in the *Yes* class. This is due to the skewness of the weak labels distribution, which is mainly composed of *Yes* labels. The distribution of Yes/No labels towards with regard to each question is visible in 4.

Finally, within each task the F1-scores are pretty similar between each of the classes: .69 and .70 for the SAC task and .97 and .93 for the IAD task.

**Validation by an Expert**   Here we analyze how accurate were the NLLF generated by the BERT-like model, and also the weak labels by the LLM. We took 100 examples from the validation set used to train the NLLFG, and asked an expert to manually label them regarding the labels of a BSQ. We compare the labeling of the expert with the outputs of the NLLFG and ChatGPT models, using classical classification metrics such as precision, recall and F1-score. We focus only on the SAC task has we just saw earlier that is the most challenging for the NLLFG during its training.

The results for both the tasks are available in Table 3. The LLM obtain a better F1-score than the smaller transformer model, which was expected. It is interesting to note that the accuracy of the NLLFG model is of 0.68, which is not its accuracy on the weak labels validation set times the accuracy of the LLM ($0.70 \times 0.78$) that is 0.55. This suggests that the NLLFG compensate some errors of the LLM.

Another important details regarding the propagation of the NLLF errors in the tree, is that the tree

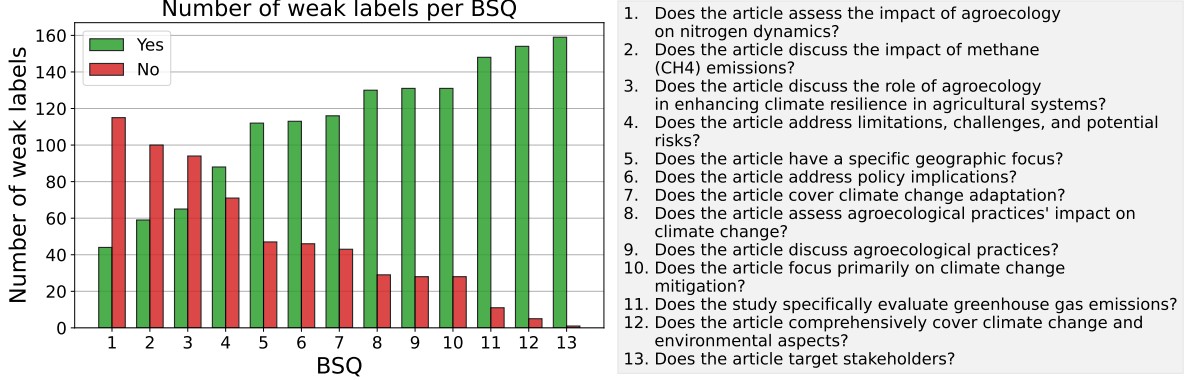

Figure 4: Weak label distributions (Yes/No) for each binary-subtask question of the SAC task.

is not using as input a class label but the probability of each label, which contains more information.

| Model | Label | Prec. | Rec. | F1 | Acc. |
|---|---|---|---|---|---|
| ChatGPT | Yes | 71 | 89 | 79 | 78 |
| | No | 88 | 68 | 77 | |
| NLLFG | Yes | 60 | 96 | 74 | 68 |
| | No | 92 | 43 | 59 | |

Table 3: NLLFG and ChatGPT performances on a set of 100 examples annotated manually by an expert.

## 4.2 Features

**Decision Tree Selected Features**   The DTs combining distinct features (see last three rows of Table 1) are free to select the features they deem best to solve the main decision-making task. For the IAD task, the decision tree combining NLLF + EF considers 25 features of which 14 are NLLF and 11 are EF. An exemplar is shown in Figure 15 on the Appendix, where the tree starts by checking for less than 3 tokens and the presence of pronouns. Otherwise, it checks whether the answer provides evidence or reasons, vowels, binary words, use of calculations, and adequate knowledge of the topic raised in the question. For the SAC task, the decision tree combining NLLF + EF considers 22 features of which 14 are NLLF and 8 are EF.

**Correlation with Main Tasks Labels**   In the IAD task, we observed that some classes correlated with some EFs more strongly than with NLLFs, due to meticulous feature design versus intuitive BSQ design, respectively. However, when looking at the results of the SAC task, it is possible to validate that our approach is functioning even to tasks that lack a known powerful, hand-crafted feature

design, but just with some keywords spotting using regular expressions.

**Causality**   In addition to provide interpretability, our method tends to foster causal learning. Indeed, by allowing the user to directly write the features to use as input, this method prevents the model to rely too heavily on latent correlational patterns that are specifically associated to certain classes (Gilpin et al., 2018; Angwin et al., 2016). Nevertheless, feature selection still relies on data distribution which makes the system not completely causal even though it tends to.

## 4.3 Path visualization

The possible path are visible in Figure 15 for the IAD task and in Figure 16 for the SAC task. All the possible paths are composed of mixed type of features with EF and NLLF.

For the SAC task, we can see that the first decision derives from the answer to "*Does the abstract address the relationship between agroecological practices and climate change?*" which allow to coarsely separate the samples between the two classes. Then, if the prefix "*convent*" is contained in the text, it is 5.6 times more probable (158/28) that the text comes from include class according to the GINI value. Examples of decisions with their associated paths in the tree are shown in Figures 13 and 14.

## 5   Conclusion and Future Work

We proposed a new method to leverage low-level reasoning knowledge related to a more complex task from a large language model, and integrate this into a smaller transformer. The transformer has been trained in a way that it can be used for zero-shot inference with any low-level reasoning having

the task formulated in natural language. This allows the practitioners to formulate easily their own features related about the task. The Natural Language Learned Feature vector can then be used as a representation in any other classifier. We show that it is easy to train an interpretable model like a Decision Tree, leading to both competitive results and interpretability. Our method can be applied to any predictive task using text as input. Future work should focus on investigating the potential impacts of this approach in real-world educational settings, and especially inspect the preferences of the practitioners regarding different explainability methods in order to help them taking complex decision (Jesus et al., 2021).

## Limitations

Although, on paper, our method is universal, we need to show that our results can generalize to other tasks where LLM (such as ChatGPT) struggle in their reasoning process, e.g., theory of mind tasks (Ullman, 2023). Moreover, our approach has been demonstrated on binary classification, and it remains to demonstrate that our approach can scale well to more categories. Otherwise, more complex tasks like multi-hop reasoning would be a late target for our system, as a simple classifier cannot solve this as it is, which would require many adaptations.

Despite the fact that we obtained promising results with our approach, the performance of both BERT and the decision tree using NLLF alone was not exceptional. This may be affected by the performance of NLLFG. In particular, we used a limited set of examples to train our NLLFG. It is possible that training with a large set of answers and BSQs, or using a prompt-based approach (Schick and Schütze, 2022) useful in few-shot setting, may improve the results. Especially, we have also seen during our experiments that the number of examples shown to the NLLFG during its training was correlated with its performances, for this reason we would like to monitor the performances of the NLLFG when trained with more weak labels from the LLM.

Finally, we claim that choosing the features fosters causality but without rigorous experimentation. This could be tested by using a dataset with a deliberately inflated bias (Reif and Schwartz, 2023).

## Ethics Statement

This work is in compliance with the ACL Ethics Policy as it allows to create models that might be more interpretable and more causal.

## Acknowledgments

The authors would like to thank Angelica Marchetti for her useful help on creating the expert features for the SAC dataset, and for the manual assessment of the weak labels in Table 3. This work was funded by National Center for Artificial Intelligence CENIA FB210017, Basal ANID.

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

## A  NLI-like Training

The BSQ (a binary subtask question in natural language) were integrated inside the transformer input as follow:

[CLS] `Input text` [SEP] `BSQ` [SEP] .

We used a pre-trained BERT (resp. BETO) model in English (resp. Spanish) language for the SAC (resp. IAD) task. We fine-tuned one only model for all the subtasks, by integrating the subtask as string and using the binary low-level subtasks labels *Yes* or *No* as *Entailment* and *Contradiction* in NLI. We trained the model for 7 epochs, using a batch size of 16 and the Cross Entropy loss, with the Adam optimizer and a learning rate of $8 \times 10^{-5}$.

## B  Decision Tree Training

we used the gini impurity as criterion to optimized and fixed the maximum depth of the tree to 5 in order to keep it comprehensible for the practitioners.

During the learning phase, we used the Gini impurity as the criterion to optimized and fixed the maximum depth of the tree to 5 in order to keep it comprehensible for the practitioners. For the the model with BoNG only, we augmented the maximum depth to 10 because of the sparsity of the features. The model with NLLF-BoNG had a minimum impurity of $1.2 \times 10^{-3}$, while the other models had a minimum impurity decrease of zero.

The BoNG were also implemented using scikit-learn, we choose 1000 as the number max of feature in order to keep the dimension small and computed the tf-idf for each n-gram.

## C  BERT fine-tuning

We used 8 epochs, a batch size of 32 and the Cross Entropy loss, with the Adam optimizer and a learning rate of $1 \times 10^{-5}$, and $5 \times 10^{-6}$ for the augmented transformers because of the concatenation of high-level features before the output layer. We selected the best model on the validation set using best accuracy for SAC and best loss for IAD (because the

| Model | Label | Prec. | Rec. | F1 | Acc. |
|-------|-------|-------|------|-----|------|
| ChatGPT | Yes | 71 | 89 | 79 | 78 |
|         | No  | 88 | 68 | 77 |      |
| NLLFG | Yes | 60 | 96 | 74 | 68 |
|       | No  | 92 | 43 | 59 |      |
| CoT | Yes | 78 | 85 | 81 | 79 |
|     | No  | 81 | 72 | 76 |      |

Table 4: NLLFG, ChatGPT and CoT performances on a set of 100 examples annotated manually by an expert.

target metric focused on the minority class, accuracy was deemed less pertinent).

## D  Chain-of-Thought for Weak Labels

We used zero-shot CoT (Kojima et al., 2022b) to generate the weak labels. As shown in the Table 4, it allows to slightly improve the results, but as we tested over 100 manually labeled examples only this is not significant. Using OpenAI API comes with a cost, as the prompts and the outputs are longer (in average 0.104 USD more per thousand queries).

## E  Prompt Templates

We show in Figures 5 and 6 the prompts used by the LLM in order to create the BSQ and the associated label for respectively the IAD and the SAC tasks.

The prompts to train the LLM to tackle the IAD-task are in Figures 7 and 8, for Vanilla and CoT, respectively. We also insert in the prompt the definition of coherence we gave to the annotators. This process did not change the results.

On the other hand, the prompts to train the LLM to tackle the SAC task are in Figures 10 and 11, for Vanilla and CoT, respectively.

## F  Binary Questions

We show in Table 6 the $N$ binary questions that were obtained by several means, before the feature selection process for the IAD task; and Tables 7 and 8 for the SAC task.

## G  Expert Features

We show in Table 9 the expert features by three categories, called Traditional, Semantic and Contextual features for the IAD task. On the other side, We show in Table 10 the expert features by two categories, called Keyword and Prefix features for the SAC task.

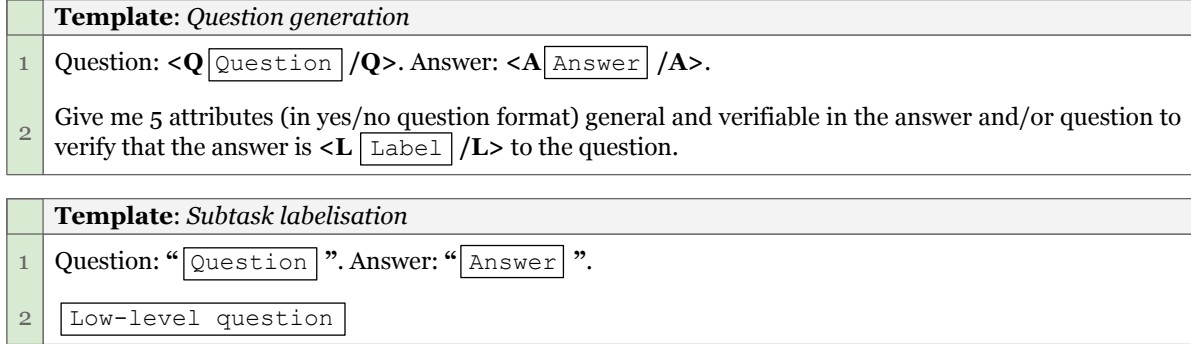

| | **Template**: *Question generation* |
|---|---|
| 1 | Question: **<Q** `Question` **/Q>**. Answer: **<A** `Answer` **/A>**. |
| 2 | Give me 5 attributes (in yes/no question format) general and verifiable in the answer and/or question to verify that the answer is **<L** `Label` **/L>** to the question. |

| | **Template**: *Subtask labelisation* |
|---|---|
| 1 | Question: " `Question` ". Answer: " `Answer` ". |
| 2 | `Low-level question` |

Figure 5: Prompts used to generate the subtasks binary questions, and create the labels regarding the subtasks questions for the IAD task, using a LLM. Translated from Spanish.

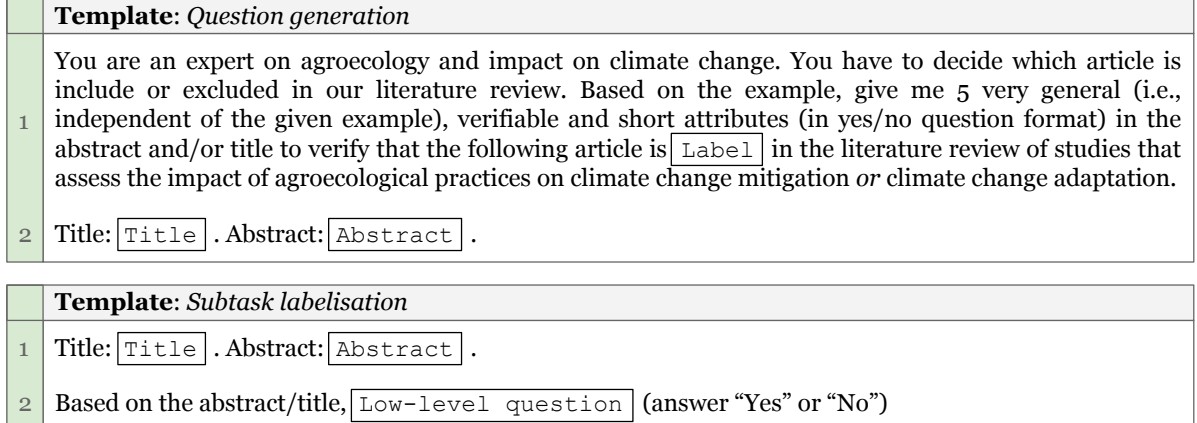

| | **Template**: *Question generation* |
|---|---|
| 1 | You are an expert on agroecology and impact on climate change. You have to decide which article is include or excluded in our literature review. Based on the example, give me 5 very general (i.e., independent of the given example), verifiable and short attributes (in yes/no question format) in the abstract and/or title to verify that the following article is `Label` in the literature review of studies that assess the impact of agroecological practices on climate change mitigation *or* climate change adaptation. |
| 2 | Title: `Title` . Abstract: `Abstract` . |

| | **Template**: *Subtask labelisation* |
|---|---|
| 1 | Title: `Title` . Abstract: `Abstract` . |
| 2 | Based on the abstract/title, `Low-level question` (answer "Yes" or "No") |

Figure 6: Prompts used to generate the subtasks binary questions, and create the labels regarding the subtasks questions for the SAC task, using a LLM.

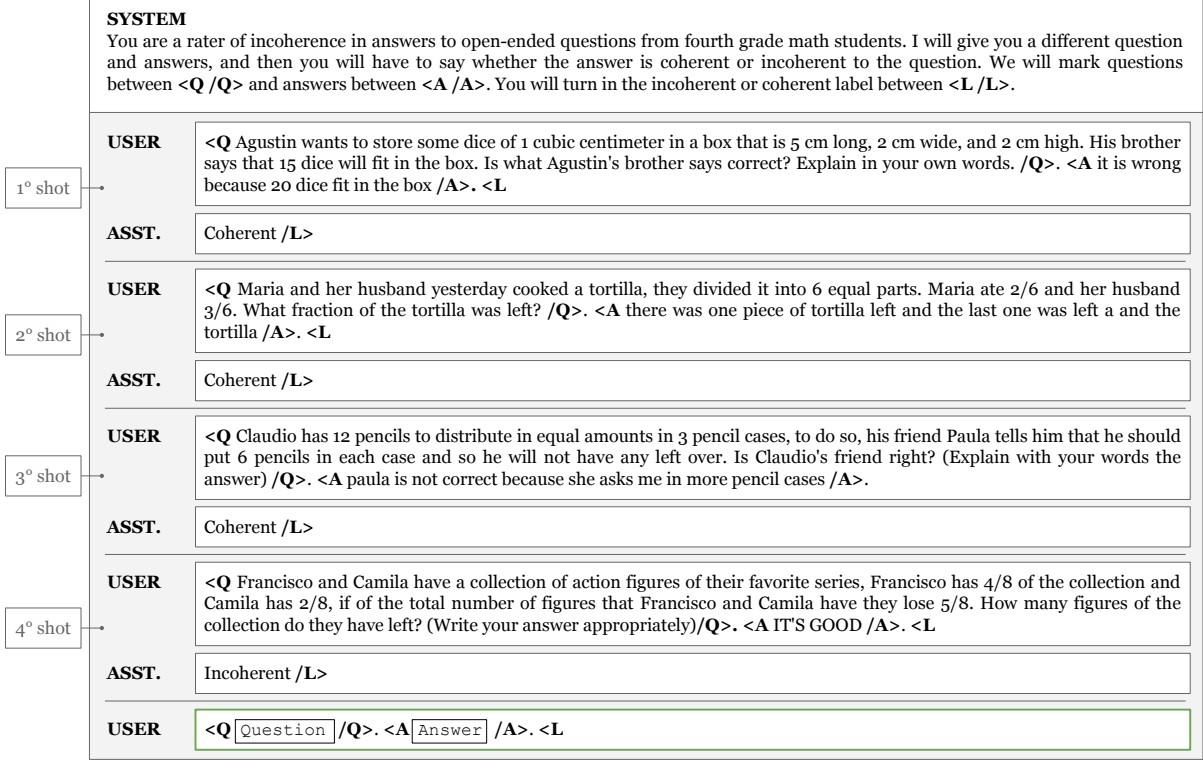

Figure 7: Vanilla ChatGPT prompt templates for the IAD task. System/User/Assistant (asst.) are the roles in the ChatGPT API. Translated from Spanish.

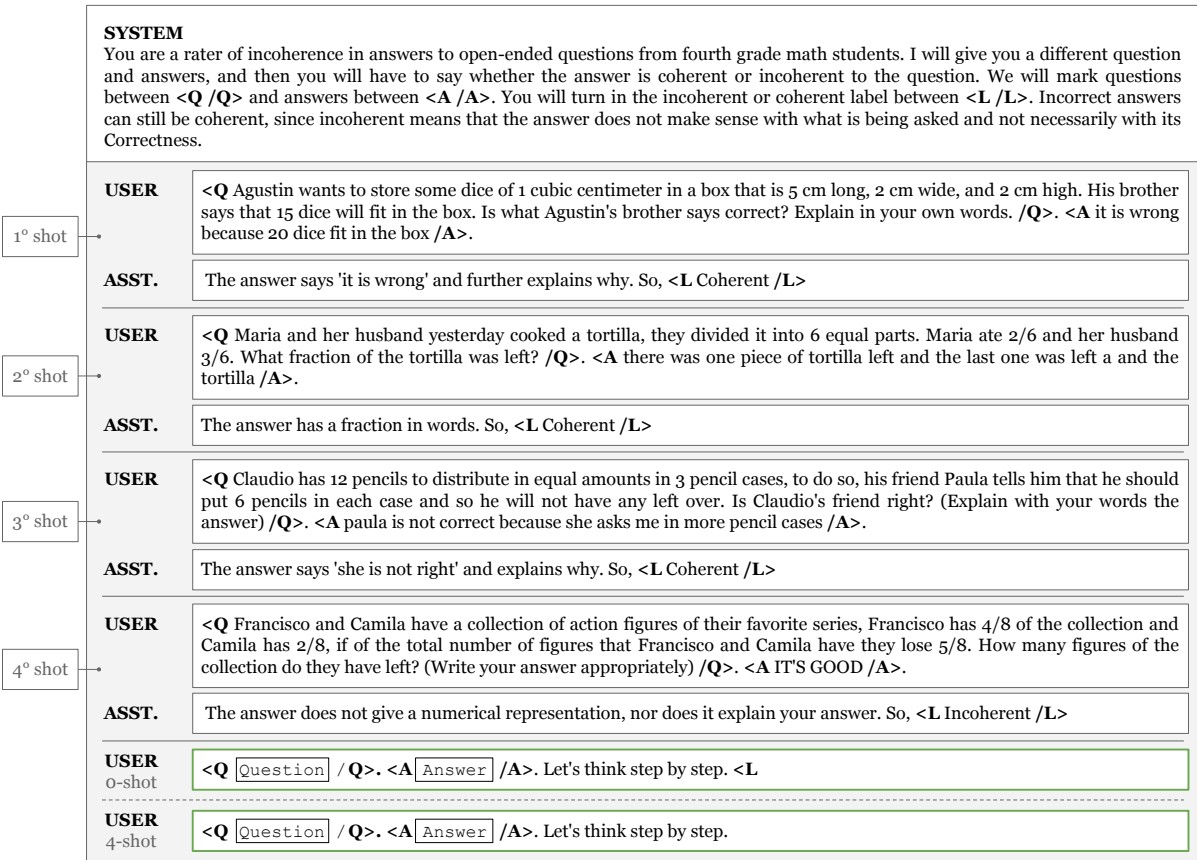

Figure 8: CoT ChatGPT prompt templates for the IAD task. System/User/Assistant (asst.) are the roles in the ChatGPT API. The phrase under the User role indicates which template is used in 0-shot or 4-shot format. Translated from Spanish.

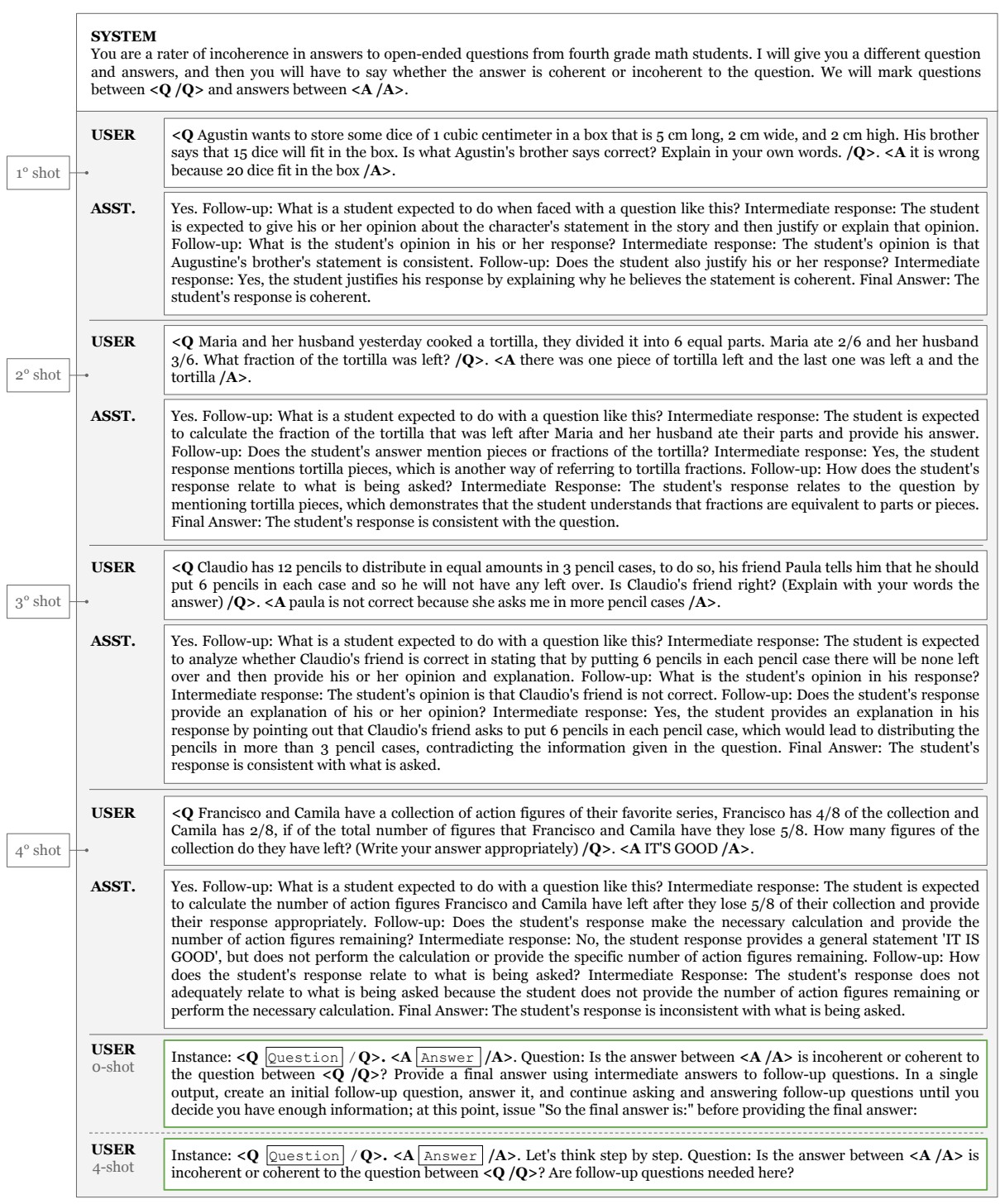

**SYSTEM**
You are a rater of incoherence in answers to open-ended questions from fourth grade math students. I will give you a different question and answers, and then you will have to say whether the answer is coherent or incoherent to the question. We will mark questions between **<Q /Q>** and answers between **<A /A>**.

**1° shot**

**USER**
**<Q** Agustin wants to store some dice of 1 cubic centimeter in a box that is 5 cm long, 2 cm wide, and 2 cm high. His brother says that 15 dice will fit in the box. Is what Agustin's brother says correct? Explain in your own words. **/Q>**. **<A** it is wrong because 20 dice fit in the box **/A>**.

**ASST.**
Yes. Follow-up: What is a student expected to do when faced with a question like this? Intermediate response: The student is expected to give his or her opinion about the character's statement in the story and then justify or explain that opinion. Follow-up: What is the student's opinion in his or her response? Intermediate response: The student's opinion is that Augustine's brother's statement is consistent. Follow-up: Does the student also justify his or her response? Intermediate response: Yes, the student justifies his response by explaining why he believes the statement is coherent. Final Answer: The student's response is coherent.

**2° shot**

**USER**
**<Q** Maria and her husband yesterday cooked a tortilla, they divided it into 6 equal parts. Maria ate 2/6 and her husband 3/6. What fraction of the tortilla was left? **/Q>**. **<A** there was one piece of tortilla left and the last one was left a and the tortilla **/A>**.

**ASST.**
Yes. Follow-up: What is a student expected to do with a question like this? Intermediate response: The student is expected to calculate the fraction of the tortilla that was left after Maria and her husband ate their parts and provide his answer. Follow-up: Does the student's answer mention pieces or fractions of the tortilla? Intermediate response: Yes, the student response mentions tortilla pieces, which is another way of referring to tortilla fractions. Follow-up: How does the student's response relate to what is being asked? Intermediate Response: The student's response relates to the question by mentioning tortilla pieces, which demonstrates that the student understands that fractions are equivalent to parts or pieces. Final Answer: The student's response is consistent with the question.

**3° shot**

**USER**
**<Q** Claudio has 12 pencils to distribute in equal amounts in 3 pencil cases, to do so, his friend Paula tells him that he should put 6 pencils in each case and so he will not have any left over. Is Claudio's friend right? (Explain with your words the answer) **/Q>**. **<A** paula is not correct because she asks me in more pencil cases **/A>**.

**ASST.**
Yes. Follow-up: What is a student expected to do with a question like this? Intermediate response: The student is expected to analyze whether Claudio's friend is correct in stating that by putting 6 pencils in each pencil case there will be none left over and then provide his or her opinion and explanation. Follow-up: What is the student's opinion in his response? Intermediate response: The student's opinion is that Claudio's friend is not correct. Follow-up: Does the student's response provide an explanation of his or her opinion? Intermediate response: Yes, the student provides an explanation in his response by pointing out that Claudio's friend asks to put 6 pencils in each pencil case, which would lead to distributing the pencils in more than 3 pencil cases, contradicting the information given in the question. Final Answer: The student's response is consistent with what is asked.

**4° shot**

**USER**
**<Q** Francisco and Camila have a collection of action figures of their favorite series, Francisco has 4/8 of the collection and Camila has 2/8, if of the total number of figures that Francisco and Camila have they lose 5/8. How many figures of the collection do they have left? (Write your answer appropriately) **/Q>**. **<A** IT'S GOOD **/A>**.

**ASST.**
Yes. Follow-up: What is a student expected to do with a question like this? Intermediate response: The student is expected to calculate the number of action figures Francisco and Camila have left after they lose 5/8 of their collection and provide their response appropriately. Follow-up: Does the student's response make the necessary calculation and provide the number of action figures remaining? Intermediate response: No, the student response provides a general statement 'IT IS GOOD', but does not perform the calculation or provide the specific number of action figures remaining. Follow-up: How does the student's response relate to what is being asked? Intermediate Response: The student's response does not adequately relate to what is being asked because the student does not provide the number of action figures remaining or perform the necessary calculation. Final Answer: The student's response is inconsistent with what is being asked.

**USER**
**0-shot**
Instance: **<Q** Question **/Q>**. **<A** Answer **/A>**. Question: Is the answer between **<A /A>** is incoherent or coherent to the question between **<Q /Q>**? Provide a final answer using intermediate answers to follow-up questions. In a single output, create an initial follow-up question, answer it, and continue asking and answering follow-up questions until you decide you have enough information; at this point, issue "So the final answer is:" before providing the final answer:

**USER**
**4-shot**
Instance: **<Q** Question **/Q>**. **<A** Answer **/A>**. Let's think step by step. Question: Is the answer between **<A /A>** is incoherent or coherent to the question between **<Q /Q>**? Are follow-up questions needed here?

Figure 9: Self-ask ChatGPT prompt templates for the IAD task. System/User/Assistant (asst.) are the roles in the ChatGPT API. The phrase under the User role indicates which template is used in 0-shot or 4-shot format. Translated from Spanish.

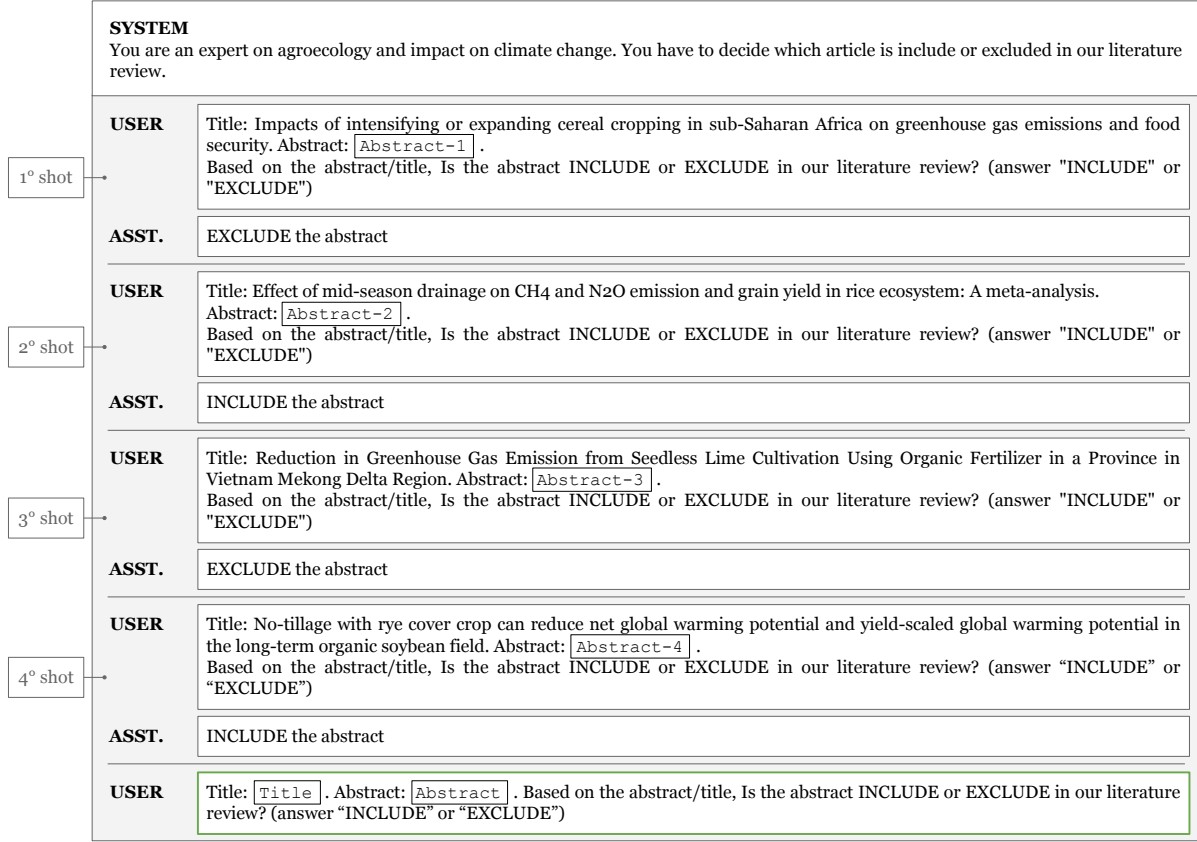

Figure 10: Vanilla ChatGPT prompt templates for the SAC task. The abstracts used are in Table 5. System/User/Assistant (asst.) are the roles in the ChatGPT API.

## H Examples of Decision Process in the Tree

We show in Figures 13 and 14 some examples along with the decision tree prediction and precision on the pathway it used.

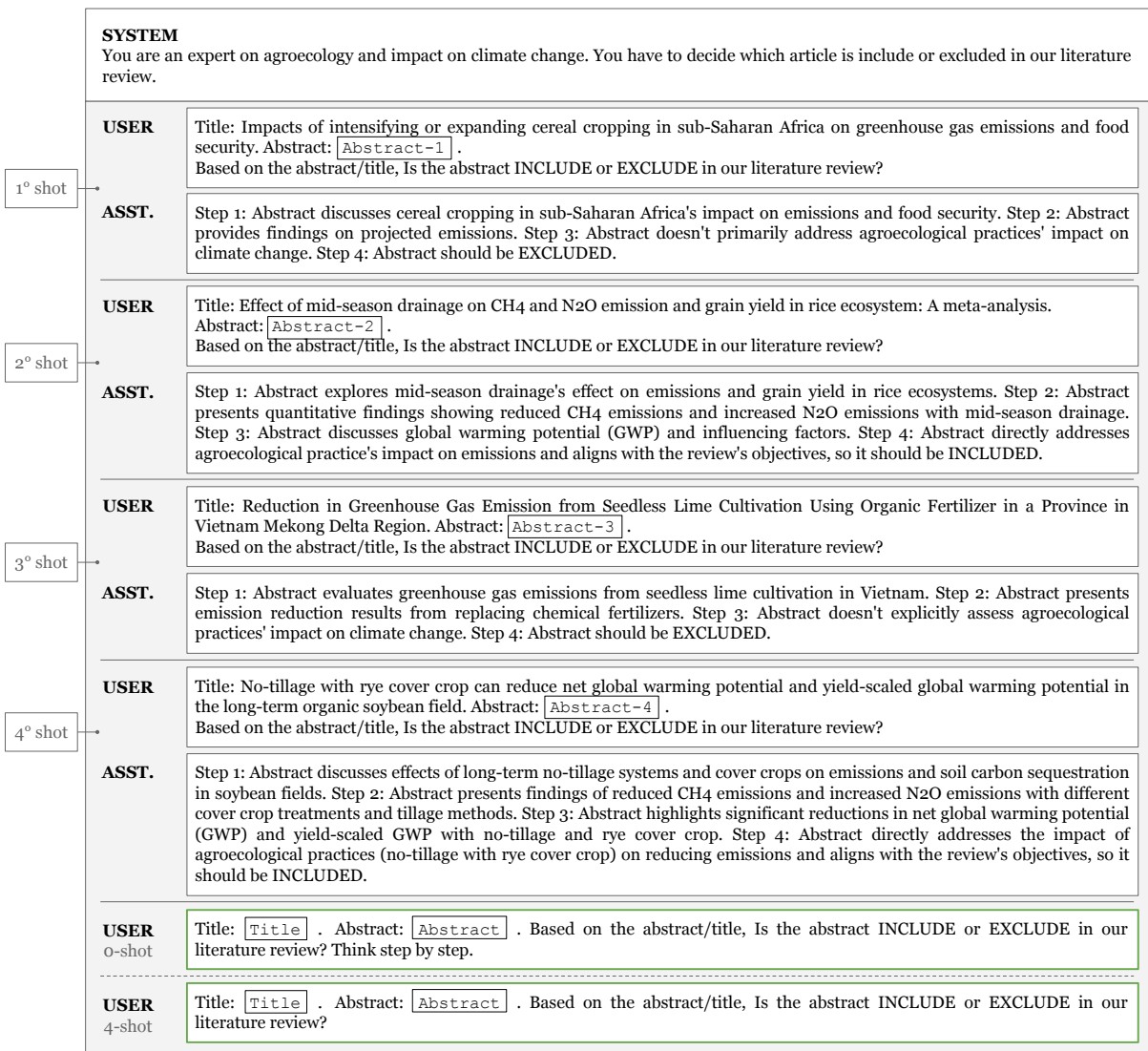

Figure 11: CoT ChatGPT prompt templates for the SAC task. The abstracts used are in Table 5. System/User/Assistant (asst.) are the roles in the ChatGPT API. The phrase under the User role indicates which template is used in 0-shot or 4-shot format.

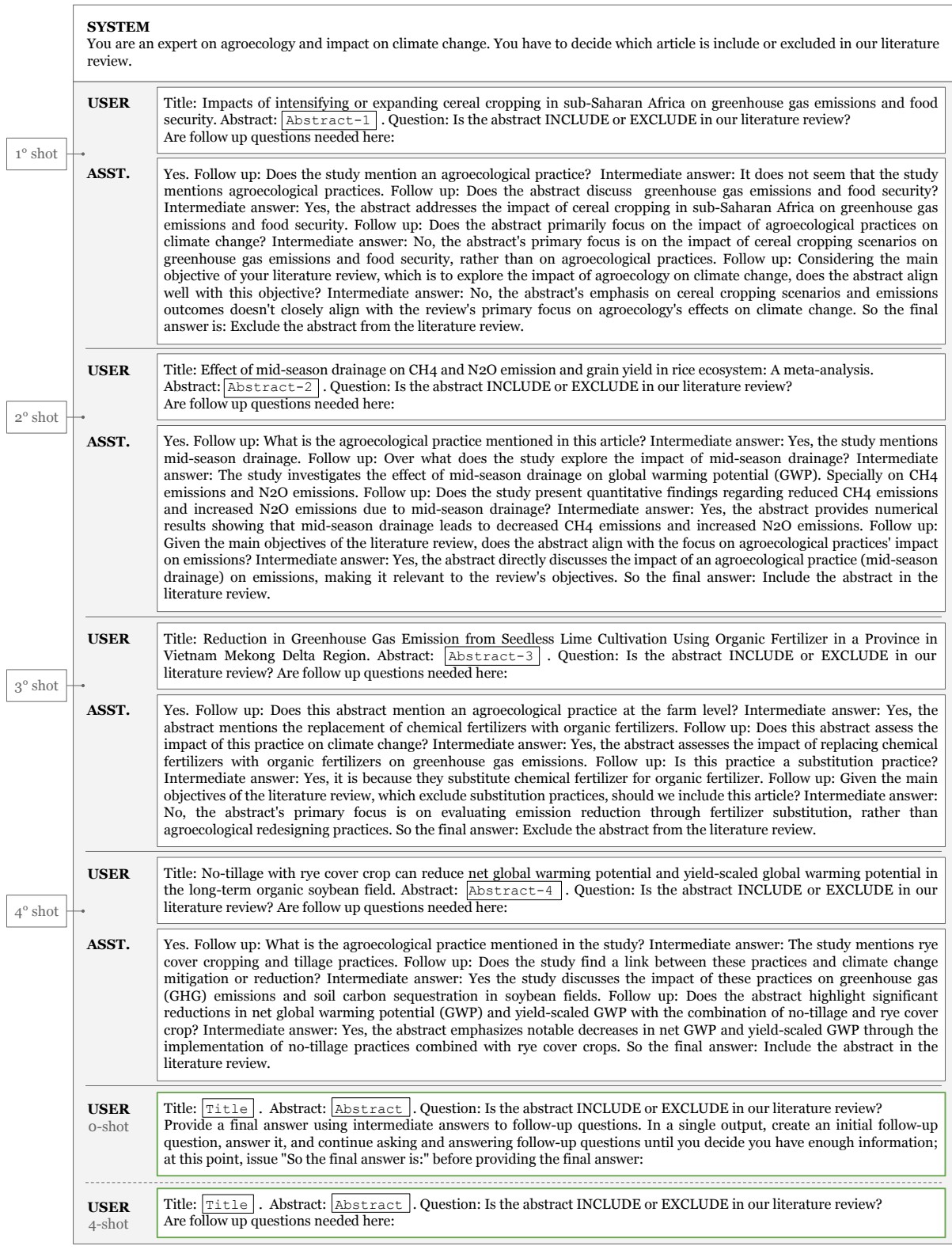

Figure 12: Self-ask ChatGPT prompt templates for the SAC task. The abstracts used are in Table 5. System/User/Assistant (asst.) are the roles in the ChatGPT API. The phrase under the User role indicates which template is used in 0-shot or 4-shot format.

| Id | Abstract |
|---|---|
| **Abstract-1** | Cropping is responsible for substantial emissions of greenhouse gasses (GHGs) worldwide through the use of fertilizers and through expansion of agricultural land and associated carbon losses. Especially in sub-Saharan Africa (SSA), GHG emissions from these processes might increase steeply in coming decades, due to tripling demand for food until 2050 to match the steep population growth. This study assesses the impact of achieving cereal self-sufficiency by the year 2050 for 10 SSA countries on GHG emissions related to different scenarios of increasing cereal production, ranging from intensifying production to agricultural area expansion. We also assessed different nutrient management variants in the intensification. Our analysis revealed that irrespective of intensification or extensification, GHG emissions of the 10 countries jointly are at least 50% higher in 2050 than in 2015. Intensification will come, depending on the nutrient use efficiency achieved, with large increases in nutrient inputs and associated GHG emissions. However, matching food demand through conversion of forest and grasslands to cereal area likely results in much higher GHG emissions. Moreover, many countries lack enough suitable land for cereal expansion to match food demand. In addition, we analysed the uncertainty in our GHG estimates and found that it is caused primarily by uncertainty in the IPCC Tier 1 coefficient for direct $N_2O$ emissions, and by the agronomic nitrogen use efficiency (N-AE). In conclusion, intensification scenarios are clearly superior to expansion scenarios in terms of climate change mitigation, but only if current N-AE is increased to levels commonly achieved in, for example, the United States, and which have been demonstrated to be feasible in some locations in SSA. As such, intensifying cereal production with good agronomy and nutrient management is essential to moderate inevitable increases in GHG emissions. Sustainably increasing crop production in SSA is therefore a daunting challenge in the coming decades |
| **Abstract-2** | Paddy rice cultivation is an important source of global anthropogenic methane emissions. Drainage the flooded soils can reduce methane substantially, but $N_2O$ emission occur concurrently, which would offset the reduction of methane emission. It remains unclear how mid-season drainage affects the global warming potential (GWP) of $CH_4$ and $N_2O$ emissions. In this study, a meta-analysis was conducted to investigate the effect of mid-season drainage on GWP and the factors that control the response of GWP to mid-season drainage. Results showed that mid-season drainage decreased $CH_4$ emission by 52% while increased $N_2O$ emission by 242%. The GWP under mid-season drainage decreased by 47% compared to continuously flooding. The yield-scaled GWP under mid-season drainage decreased by 48%. Mid-season drainage had no effect on rice grain yield. Although soil drainage times and organic matter amendment are important factors affecting $CH_4$ and $N_2O$ emissions in rice paddy field, the study showed that neither of them had effect on the response of GWP to mid-season drainage. The reduction rate of the GWP under mid-season drainage increased when N fertilization application rate increases from 50 kg ha(-1) to > 200 kg ha(-1). This study demonstrated that $CH_4$ is still a dominant greenhouse gas in rice paddies under water management with mid-season drainage. Nitrogen fertilization is an important factor that regulates the response of GWP to mid-season drainage. High nitrogen fertilization rate would decrease the overall emission of $CH_4$ and $N_2O$ under mid-season drainage. However, increasing drainage times or applying organic fertilizer under mid-season does not change the overall emission rate of $CH_4$ and $N_2O$ |
| **Abstract-3** | This study aimed to evaluate greenhouse gas (GHG) emissions from conventional cultivation (S1) of seedless lime (SL) fruit in Hau Giang province, in the Mekong Delta region of Vietnam. We adjusted the scenarios by replacing 25% and 50% of nitrogen chemical fertilizer with respective amounts of N-based organic fertilizer (S2 and S3). Face-to-face interviews were conducted to collect primary data. Life cycle assessment (LCA) methodology with the cradle to gate approach was used to estimate GHG emission based on the functional unit of one hectare of growing area and one tonnage of fresh fruit weight. The emission factors of agrochemicals, fertilizers, electricity, fuel production, and internal combustion were collected from the MiLCA software, IPCC reports, and previous studies. The S1, S2, and S3 emissions were 7590, 6703, and 5884 kg-$CO_2$ equivalent ($CO_2$e) per hectare of the growing area and 273.6, 240.3, and 209.7 kg-$CO_2$e for each tonnage of commercial fruit, respectively. Changing fertilizer-based practice from S1 to S2 and S3 mitigated 887.0-1706 kg-$CO_2$e ha(-1) (11.7-22.5%) and 33.3-63.9 kg-$CO_2$e t(-1) (12.2-25.6%), respectively. These results support a solution to reduce emissions by replacing chemical fertilizers with organic fertilizers |
| **Abstract-4** | No-tillage (NT) and the introduction of cover crops, owing to their positive effects on soil organic carbon (SOC) sequestration and crop yields, are potential agricultural practices that both support food security under the new realities of climate change and alleviate greenhouse gas (GHG) emissions. However, the effects of the combination of long-term NT systems and cover crops on non-carbon dioxide ($CO_2$) emissions and SOC sequestration have not been adequately documented, particularly in East Asia. We conducted a split-plot field experiment involving two tillage systems [NT and moldboard plowing (MP)] and three cover crops, namely, fallow (FA), hairy vetch (HV), and rye (RY). NT had slightly higher soybean yield than MP, although tillage methods and cover crop treatments had no significant effects on soybean yield. Cover crop treatments rather than tillage methods significantly affected methane ($CH_4$) emissions; under FA and RY treatments, we observed $CH_4$ uptakes, whereas under HV, we observed $CH_4$ emissions. In contrast, rather than cover crop treatments, tillage methods affected nitrous oxide ($N_2O$) emissions. Higher WFPS and soil bulk density under NT resulted in significantly higher annual $N_2O$ emissions than those under MP. However, under NT, the annual SOC sequestration rate significantly increased compared with that under MP, the global warming potential (GWP) caused by $CH_4$ and $N_2O$ emissions was fully offset by net $CO_2$ retention under NT. Additionally, treatment under NT reduced net GWP and yield-scaled GWP to a significantly greater degree than did treatment under MP. Treatments under NT with RY cover crop had the lowest net GWP (-2324 kg $CO_2$ equivalent ha(-1) year(-1)) and yield-scaled GWP (-1037 kg $CO_2$ equivalent Mg-1 soybean yield). These findings suggest that treatments under NT with cover crop systems-especially RY cover crop-in the long-term organic soybean field maintains sustainable crop production and reduces net GWP and yield-scaled GWP, which will be an effective climate-smart agriculture practice in the humid, subtropical regions prevailing in Kanto, Japan |

Table 5: Abstracts used in the prompt templates for the SAC task.

| Questions | Origin |
|---|---|
| Is the answer clear and uses appropriate language and spelling for the question asked? | LLM |
| Does the answer provide useful information without any joking, sarcastic, or ambiguous tones? | LLM |
| Does the question imply that the student should evaluate the logic and coherence of the character's answer in the story? | LLM |
| Are the processes shown to obtain the value? | LLM |
| Does the answer rule out incorrect options with justification? | LLM |
| Does the answer limit itself to a sarcastic comment instead of providing a useful and coherent answer? | LLM |
| Does the answer provide relevant and accurate information related to the question asked? | LLM |
| Is the calculation methodology present in the answer? | LLM |
| Does the answer include calculations or processes? | LLM |
| Does the answer support its position with facts? | LLM |
| Does the answer show the calculations or processes to arrive at a numerical value? | LLM |
| Does the answer appear to be a joke or a humorous answer? | LLM |
| Does the answer show a clear understanding of the mathematical concepts involved in the question? | LLM |
| Does the answer substantiate its claim with data? | LLM |
| Does the answer contain spelling or grammatical errors? | LLM |
| Does the answer not present any contradictions or ambiguities with the question asked? | LLM |
| Does the answer indicate mathematical superiority of the character? | LLM |
| Does the answer provide justification for ruling out incorrect options? | LLM |
| Does the answer consider all possible options? | LLM |
| Does the question imply deciding whether a statement is correct or incorrect? | LLM |
| Does the question present a character who must make a mathematical decision? | LLM |
| Does the answer consider all possible options and provide justification for ruling out incorrect options? | LLM |
| Does the selected character know more mathematics than others mentioned? | LLM |
| Does the answer include evidence or justification? | LLM |
| Does the answer offer arguments or examples to support its claim? | LLM |
| Does the answer provide a direct answer related to the question asked? | LLM |
| Does the answer consider all options and rule out incorrect ones? | LLM |
| Does the answer provide the calculations performed? | LLM |
| Is the selected character the most skilled in mathematics? | LLM |
| Does the answer indicate logical and coherent skills? | LLM |
| Does the answer use specific examples to support the correct character choice? | LLM |
| Does the answer demonstrate adequate knowledge and understanding of the topic raised in the question? | LLM |
| Is the answer to the question "yes" or "no"? | LLM |
| Does the answer demonstrate a logical and coherent mind? | LLM |
| Does the answer to the question imply verifying whether a mathematical operation was performed correctly? | LLM |
| Does the answer clearly indicate whether the character's statement is correct or not? | LLM |
| Does the answer to the question provide a clear explanation of why it is correct or incorrect? | LLM |
| Does the answer indicate if the selected character has more or less mathematical knowledge than other characters mentioned in the question? | LLM |
| Does the answer reflect reasoning and coherence skills? | LLM |
| Does the answer present evidence or reasons? | LLM |
| Does the answer reveal coherent thinking skills? | LLM |
| Is the answer clear, concise, and uses easy-to-understand and precise language? | LLM |
| Is a justification requested for the answer to the question? | LLM |
| Does the answer provide justification for choosing the correct character? | LLM |
| Does the answer reference other relevant sources of information that may support the correct character's choice? | LLM |
| Is the answer brief and not elaborated? | LLM |
| Does the answer reflect a clear understanding of the context and situations described in the question? | LLM |
| Does the chosen character have more mathematical skills than others? | LLM |
| Does the answer use language and spelling consistent with the question? | LLM |
| Does the answer consider all options and justify discards? | LLM |
| Does the answer demonstrate the student's ability to reason logically and follow a coherent thought process? | LLM |
| Does the answer show understanding of the question and evidence of an attempt to solve the mathematical problem posed? | LLM |
| Does the answer indicate whether the character is more mathematical than others? | LLM |
| Does the answer reveal the calculation process? | LLM |
| Does the answer show reasoning and cohesion? | LLM |
| Does the answer defend its assertion with solid arguments? | LLM |
| Does the answer make sense? | Ling |
| Does the answer contain any of the words "yes" or "no"? | Ling |
| Does the answer contradict the question? | Ling |
| Does the answer describe the process used to obtain the result or reach the conclusion? | Ling |
| Is the answer a personal opinion? | Ling |
| Does the answer involve the use of numbers or digits? | Ling |
| Does the answer have a reasonable number of tokens? | Hum |
| Does the answer have a reasonable maximum length of repeated characters? | Hum |
| Does the question suggest that something is correct or that someone is right? | Hum |
| Does the answer have a reasonable proportion of non-numeric characters? | Hum |
| Does the answer have a reasonable proportion of vowels? | Hum |
| Does the question include a proper name and is it also present in the answer? | Hum |

Table 6: Binary subtasks questions and their origin (human-made, LLM-made, or natural language translation of linguistics rules) before the feature selection process for the IAD task. Everything was translated from Spanish

| Questions | Origin |
|---|---|
| Does the abstract cover climate change adaptation? | LLM |
| Does the abstract assess the impact of agroecology on nitrogen dynamics? | LLM |
| Does the abstract address limitations, challenges, and potential risks? | LLM |
| Does the abstract discuss the role of agroecology in enhancing climate resilience in agricultural systems? | LLM |
| Does the abstract assess agroecological practices' impact on climate change? | LLM |
| Does the abstract discuss the impact of methane (CH4) emissions? | LLM |
| Does the abstract target stakeholders? | LLM |
| Does the study specifically evaluate greenhouse gas emissions? | LLM |
| Does the abstract discuss measures to mitigate climate change? | LLM |
| Does the abstract evaluate agroecology's impact on nitrogen dynamics? | LLM |
| Is agroecological practices discussed in the abstract? | LLM |
| Does the abstract touch upon policy implications? | LLM |
| Does the abstract include a comprehensive discussion on climate change and environmental aspects? | LLM |
| Does the abstract cover limitations, challenges, and potential risks? | LLM |
| Does the abstract discuss limitations, challenges, and potential risks? | LLM |
| Does the abstract address methane (CH4) emissions' impact? | LLM |
| Does the abstract thoroughly address climate change and environmental aspects? | LLM |
| Does the abstract examine the implications of methane (CH4) emissions? | LLM |
| Does the abstract mention limitations, challenges, and potential risks? | LLM |
| Does the abstract provide a comprehensive coverage of climate change and environmental aspects? | LLM |
| Does the abstract discuss policy implications? | LLM |
| Does the abstract consider policy implications? | LLM |
| Does the abstract analyze how agroecology affects nitrogen dynamics? | LLM |
| Does the abstract examine the impact of agroecology on nitrogen dynamics? | LLM |
| Does the abstract analyze the role of agroecology in carbon sequestration? | LLM |
| Does the abstract discuss agroecology's benefits for biodiversity? | LLM |
| Does the abstract examine the impact of these practices? | LLM |
| Does the abstract lack empirical evidence or scientific research? | LLM |
| Does the abstract suggest a lack of correlation? | LLM |
| Does the abstract primarily address peat emissions and their quantification? | LLM |
| Do the statistical analyses support the findings? | LLM |
| Does the abstract explore soil organic sequestration rate? | LLM |
| Does the abstract compare organic and conventional arable farming practices? | LLM |
| Does the abstract discuss agroecology's benefits for ecosystem services? | LLM |
| Does the abstract discuss cultural aspects of agroecology? | LLM |
| Does the abstract focus primarily on economic aspects of agroecology? | LLM |
| Does the abstract discuss biodiversity conservation? | LLM |
| Does the abstract review previous studies? | LLM |
| Does the abstract provide evidence of the impact? | LLM |
| Does the abstract discuss social aspects of agroecology? | LLM |
| Does the abstract mention optimized timing of grass-clover ley phase removal? | LLM |
| Does the abstract discuss N2O emissions? | LLM |
| Does the abstract focus on NH3 fluxes? | LLM |
| Does the abstract discuss soil health? | LLM |
| Does the abstract compare industrial agriculture practices? | LLM |
| Does the abstract discuss certified organic production? | LLM |
| Does the abstract specifically examine the impact on GHG profiles? | LLM |
| Does the abstract evaluate rubber-leguminous shrub systems? | LLM |
| Does the abstract focus on small-scale or family farming systems? | LLM |
| Does the abstract specifically analyze nitrous oxide emissions? | LLM |
| Is there evidence of NH3 and GHG fluxes? | LLM |
| Does the abstract measure field plots? | LLM |
| Does the abstract analyze yield-scaled global warming potential? | LLM |
| Does the study focus on conventional cultivation methods? | LLM |
| Does the abstract examine nitrogen dynamics? | LLM |
| Does the abstract provide recommendations? | LLM |
| Does the abstract discuss biofuel production? | LLM |
| Does the abstract offer proof for its conclusions? | LLM |
| Does the abstract solely focus on cradle-to-farm-gate activities? | LLM |
| Does the abstract measure net global warming potential? | LLM |
| Does the abstract emphasize the United States? | LLM |
| Does the abstract lack any new empirical data or fresh insights? | LLM |
| Does the abstract primarily emphasize economic aspects of agroecology? | LLM |
| Does the abstract specifically address GHG profiles in its examination? | LLM |
| Is the main focus of the abstract on the economic aspects of agroecology? | LLM |
| Does the abstract center around the economic aspects of agroecology? | LLM |
| Does the abstract specifically assess the effects of these practices? | LLM |
| Does the abstract cover agroecology's positive impact on biodiversity? | LLM |
| Does the abstract cover biofuel production? | LLM |
| Does the abstract address the benefits of agroecology for biodiversity? | LLM |
| Does the abstract examine the specific impact of these practices? | LLM |
| Does the abstract examine how agroecology benefits biodiversity? | LLM |

Table 7: Binary subtasks questions and their origin (LLM-made) before the feature selection process for the SAC task.

| Questions | Origin |
|---|---|
| Does the abstract mention any terms starting with 'bio'? | Ling |
| Does the abstract specifically mention CH4? | Ling |
| Does the abstract discuss emissions? | Ling |
| Does the abstract discuss reducing something? | Ling |
| Does the abstract make reference to the concept of cover? | Ling |
| Is the concept of intercropping mentioned in the abstract? | Ling |
| Does the abstract discuss strategies? | Ling |
| Does the abstract address the topic of GHG emissions? | Ling |
| Does the abstract refer to the application of a type of organic fertilisation practice? | Hum |
| Does the abstract refer to the impact (or effect) of these practices on Nitrogen/N2O/nitrogen oxide emissions? | Hum |
| Does the abstract refer to the impact (or effect) of these practices on the carbon sequestration in the soil? | Hum |
| Does the abstract refer to the impact (or effect) of these practices on Carbon/CH4/methane emissions? | Hum |
| Does the abstract refer to the application of one or more Climate-Smart Agriculture practices? | Hum |
| Does the abstract refer to the impact (or effect) of these practices on climate change mitigation? | Hum |
| Does the abstract refer to the application of one or more Sustainable Rice Intensification practices? | Hum |
| Does the abstract refer to the impact (or effect) of these practices on climate change adaptation? | Hum |
| Does the abstract refer to the impact (or effect) of these practices on greenshouse gasses (GHG) emissions? | Hum |
| Does the abstract refer to the application of a type of Bio-control practice? | Hum |
| Does the abstract refer to the application of one or more agroecological practices? | Hum |
| Does the abstract refer to the application of a type of ecological or mechanical weed management practice? | Hum |
| Does the abstract refer to the application of one or more Diversified farming practices? | Hum |
| Does the abstract evaluate agroecological practices' impact on climate change? | Hum |
| Does the abstract mention any organic agriculture practices being applied? | Hum |
| Does the abstract discuss the substitution of different varieties or cultivars? | Hum |
| Does the abstract explore the connection between agroecological practices and climate change? | Hum |
| Does the abstract analyze how agroecological systems affect climate change? | Hum |
| Does the abstract mention the replacement of various varieties or cultivars? | Hum |
| Does the abstract mention the implementation of Sustainable Rice Intensification practices? | Hum |
| Does the abstract address how these practices affect soil carbon storage? | Hum |
| Does the abstract discuss the application of Regenerative agriculture methods? | Hum |
| Does the abstract discuss a form of Residues management practice? | Hum |
| Does the abstract mention the use of intercropping practices? | Hum |
| Does the abstract mention the use of Regenerative agriculture practices? | Hum |
| Do the contents of the abstract pertain to agroecological practices? | Hum |
| Is the abstract discussing the application of agroecological methods? | Hum |
| Does the abstract mention the application of cover crops or mulching? | Hum |
| Does the abstract discuss implementing a type of water collection practice? | Hum |

Table 8: Binary subtasks questions and their origin (human-made or natural language translation of linguistics rules) before the feature selection process for the SAC task.

| Expert Feature | Category |
|---|---|
| The answer is blank | Traditional |
| Number of non-space characters in the response | Traditional |
| Proportion of punctuation symbols in the response | Traditional |
| Proportion of non-vowel punctuation symbols in the response | Traditional |
| Proportion of vowels in the response | Traditional |
| Number of valid words in the response | Traditional |
| Maximum length of consecutive characters in the response | Traditional |
| Proportion of characters that are digits in the response | Traditional |
| Maximum length of consecutive vowel symbols in the response | Traditional |
| Proportion of symbols that are not digits or non-mathematical punctuation symbols in the response | Traditional |
| Number of alphabetical symbols in the response | Traditional |
| Proportion of symbols that are vowels in the response | Traditional |
| Number of words in the response | Traditional |
| Length of the response | Traditional |
| Number of numerical representations in the response | Traditional |
| Number of mathematical punctuation symbols in the response | Traditional |
| Number of digits in the response | Traditional |
| Number of punctuation symbols in the response | Traditional |
| Number of non-number words in the response | Traditional |
| Proportion of symbols that are vowels in the response | Traditional |
| Proportion of symbols that are not numbers in the response | Traditional |
| Proportion of punctuation symbols in the response | Traditional |
| There is a numerical representation (integer, real, fraction) in the response | Traditional |
| Number of symbols in the longest number in the response | Traditional |
| Proportion of punctuation symbols or digits in the response | Traditional |
| Proportion of non-mathematical punctuation symbols in the response | Traditional |
| Maximum length of consecutive non-vowel symbols in the response | Traditional |
| The answer is a digit | Traditional |
| Frequency of the letter "k" in the response | Traditional |
| Frequency of the letter "g" in the response | Traditional |
| Frequency of the letter "y" in the response | Traditional |
| Frequency of the letter "j" in the response | Traditional |
| Frequency of the letter "h" in the response | Traditional |
| Frequency of the letter "x" in the response | Traditional |
| Frequency of the letter "w" in the response | Traditional |
| Frequency of the letter "ñ" in the response | Traditional |
| The answer is an emoticon | Semantic |
| The answer is "I don't know" | Semantic |
| The answer is a greeting | Semantic |
| The answer contains offensive language | Semantic |
| The answer contains emoticons | Semantic |
| Proportion of non-emoticon faces in the response | Semantic |
| Proportion of keywords in the response | Semantic |
| Number of emoticons in the response | Semantic |
| Number of words in the RAE (Real Academia Española) in the response | Semantic |
| Number of words in the Urban Dictionary in the response | Semantic |
| Number of popular words in the response | Semantic |
| Number of keywords in the response | Semantic |
| Proportion of words in the response that are in the Royal Spanish Academy dictionary | Semantic |
| Proportion of words in the response that are in an urban dictionary | Semantic |
| Proportion of popular words in the response | Semantic |
| Proportion of keywords in the response | Semantic |
| Proportion of emoticons in the response | Semantic |
| Intersection between words in the response and words in the question that are nominal subjects | Contextual |
| The question asks for which, who or what | Contextual |
| The question asks if something is possible | Contextual |
| The answer is binary, yes or no | Contextual |
| The question asks why someone is right or wrong | Contextual |
| The question asks if something or someone is okay | Contextual |
| The answer has a reason or is binary (yes or no) | Contextual |
| Intersection between pronouns in the question and the response | Contextual |
| The question asks for who or which | Contextual |
| The question asks if someone is correct or right | Contextual |
| The question asks if someone is correct | Contextual |
| The question asks if someone is right | Contextual |
| Intersection between words in the question and the response | Contextual |

Table 9: Expert features and their category (Traditional, Semantic or Contextual) before the feature selection process for the IAD task. Everything was translated from Spanish

| Linguistic Feature | Category | Linguistic Feature | Category |
|---|---|---|---|
| intensification | Keyword | meta-analysis | Keyword |
| rainfed | Keyword | rice | Keyword |
| impact of | Keyword | CH4 | Keyword |
| net | Keyword | vineyard | Keyword |
| systems | Keyword | crop | Keyword |
| climate | Keyword | economy | Keyword |
| agricultural | Keyword | farm | Keyword |
| grass | Keyword | till | Keyword |
| soils | Keyword | practices | Keyword |
| organic | Keyword | water | Keyword |
| productivity | Keyword | methane | Keyword |
| storage | Keyword | emission | Keyword |
| scenario | Keyword | tillage | Keyword |
| farms | Keyword | conservation | Keyword |
| significantly | Keyword | seasonal | Keyword |
| cover | Keyword | social | Keyword |
| N2O | Keyword | GHG | Keyword |
| change | Keyword | agroforestry | Keyword |
| model | Keyword | potential | Keyword |
| gas | Keyword | soil | Keyword |
| strategies | Keyword | agriculture | Keyword |
| system | Keyword | experiment | Keyword |
| synthetic | Keyword | impact | Keyword |
| livestock | Keyword | greenhouse | Keyword |
| lower | Keyword | - | - |
| intercropping systems | Keyword | global warm | Prefix |
| fallow | Keyword | bio | Prefix |
| higher | Keyword | reduc | Prefix |
| predict | Keyword | emission | Prefix |
| emissions | Keyword | nitr | Prefix |
| conventional | Keyword | convent | Prefix |
| soybean | Keyword | ecolog | Prefix |
| agroforestry systems | Keyword | integrate | Prefix |
| carbon | Keyword | mitig | Prefix |
| intercropping | Keyword | increas | Prefix |

Table 10: Linguistic features and their category (Keyword and Prefix) before the feature selection process for the SAC task.

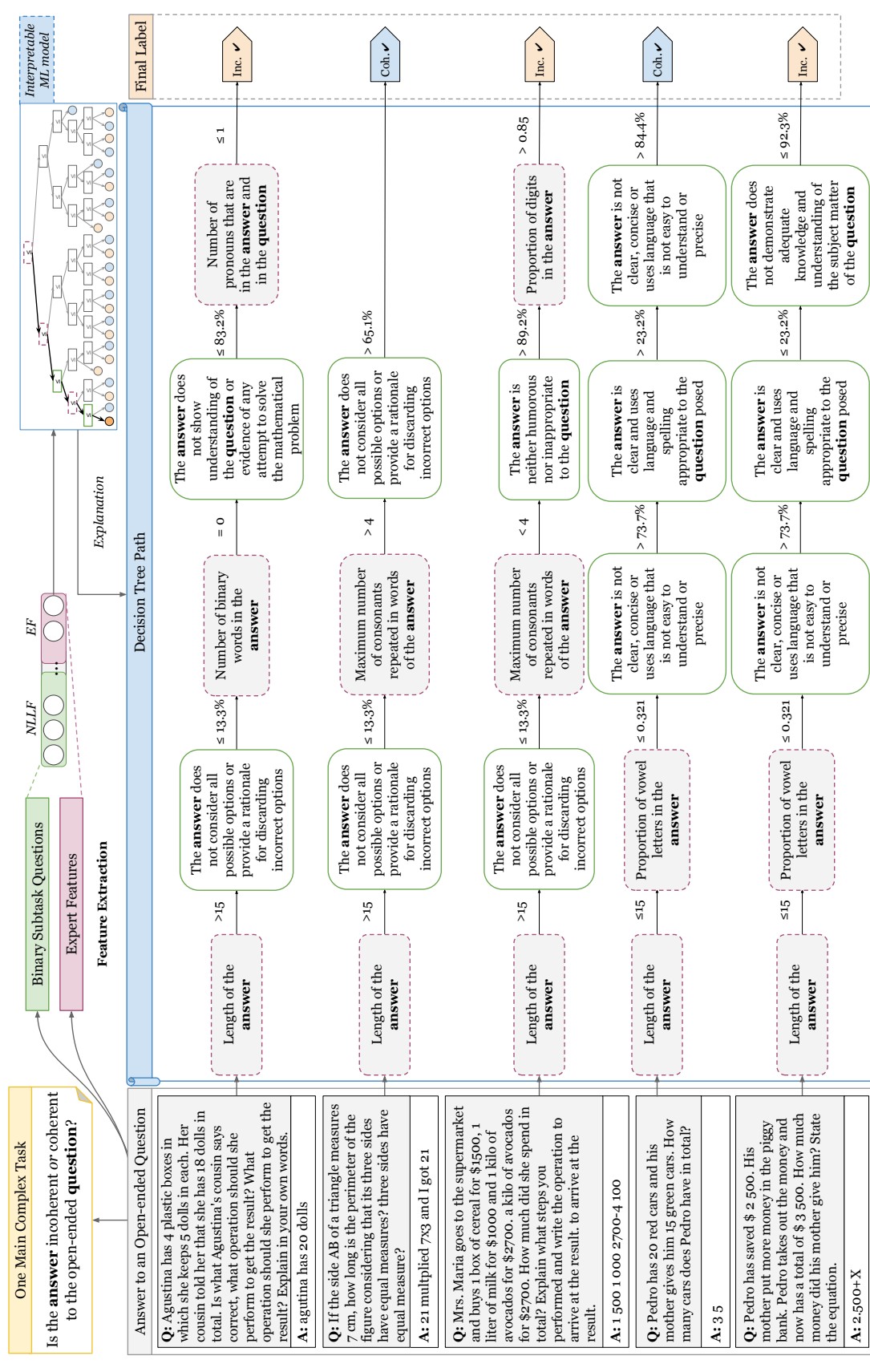

Figure 13: Explanation of the decision trees results using the features NLLF+EF for the IAD task. Inc. is incoherent, and Coh. is coherent. The check mark means that the prediction is correct. Translated from Spanish.

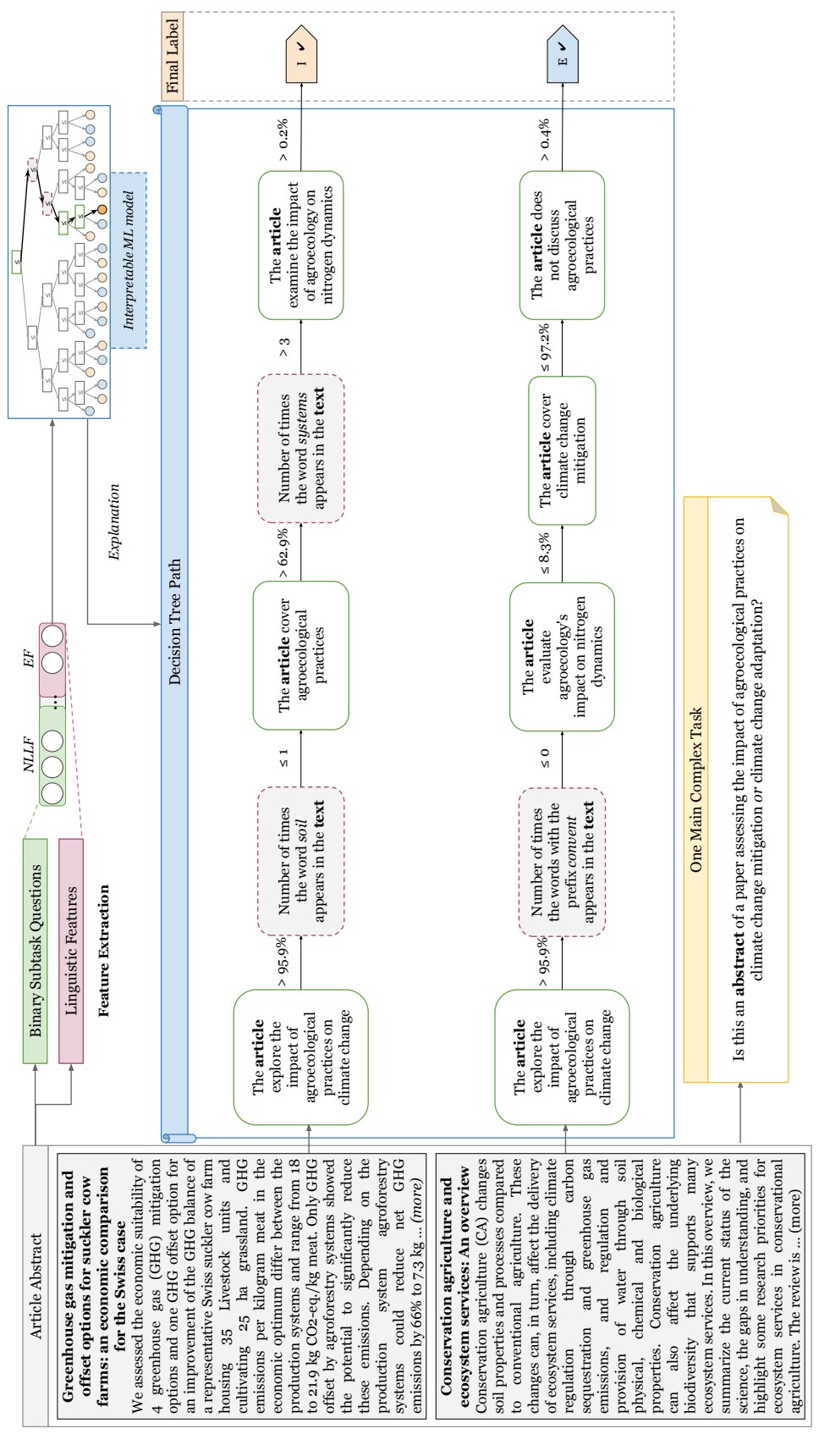

Figure 14: Explanation of the decision trees results using the features NLLF+EF for the SAC task. I is for Include, and E for Exclude. The check mark means that the prediction is correct.

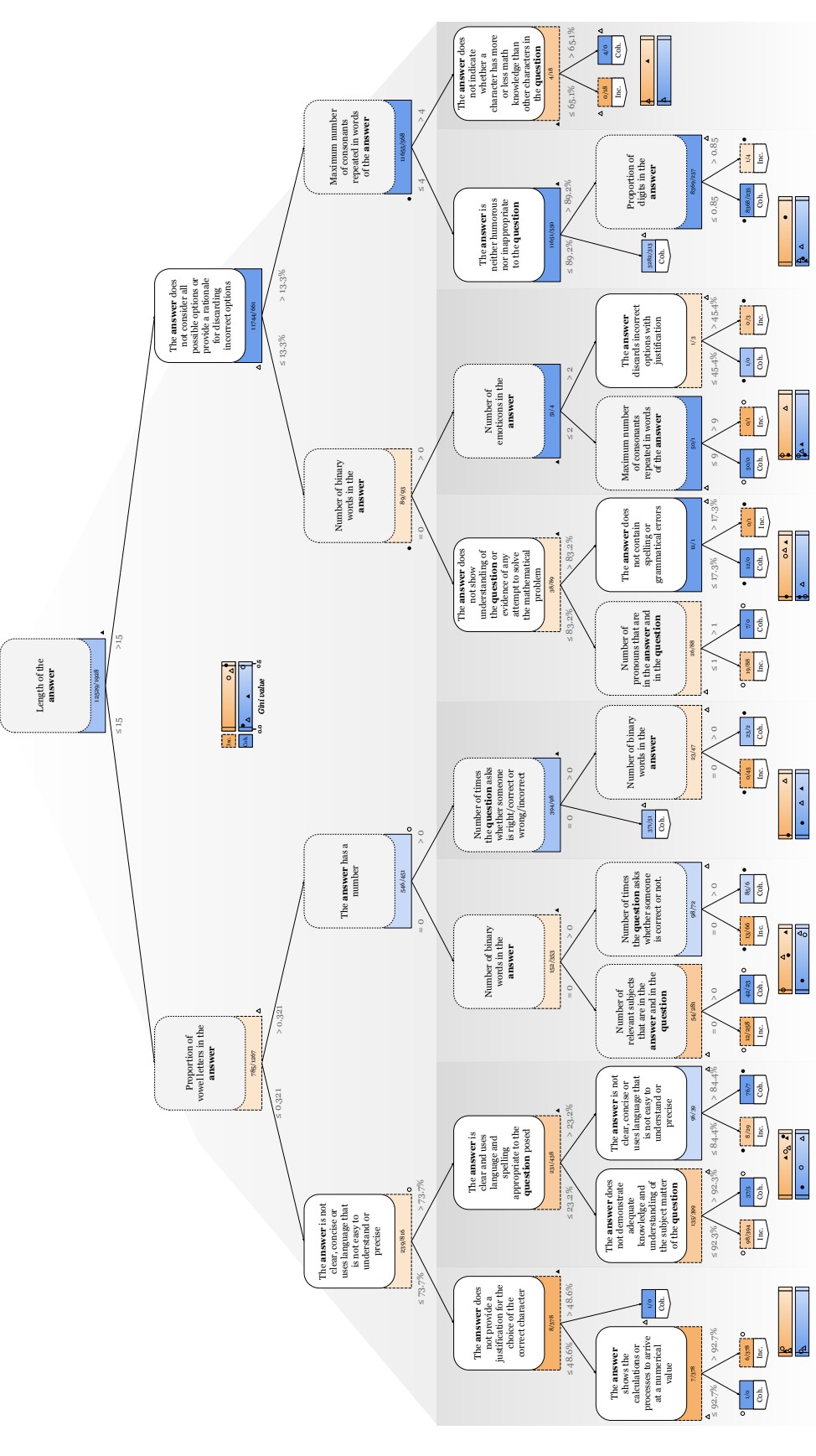

Figure 15: Decision tree with NLLF and EF for the IAD task. Translated from Spanish. Inc. is incoherent, and Coh. is coherent. The NLLF are in white and the EF are in grey. The numbers down the boxes are the threshold of the decision. The Gini value of a tree node corresponds to the ratio of the minoritarian class elements over the dominant ones in the subtree.

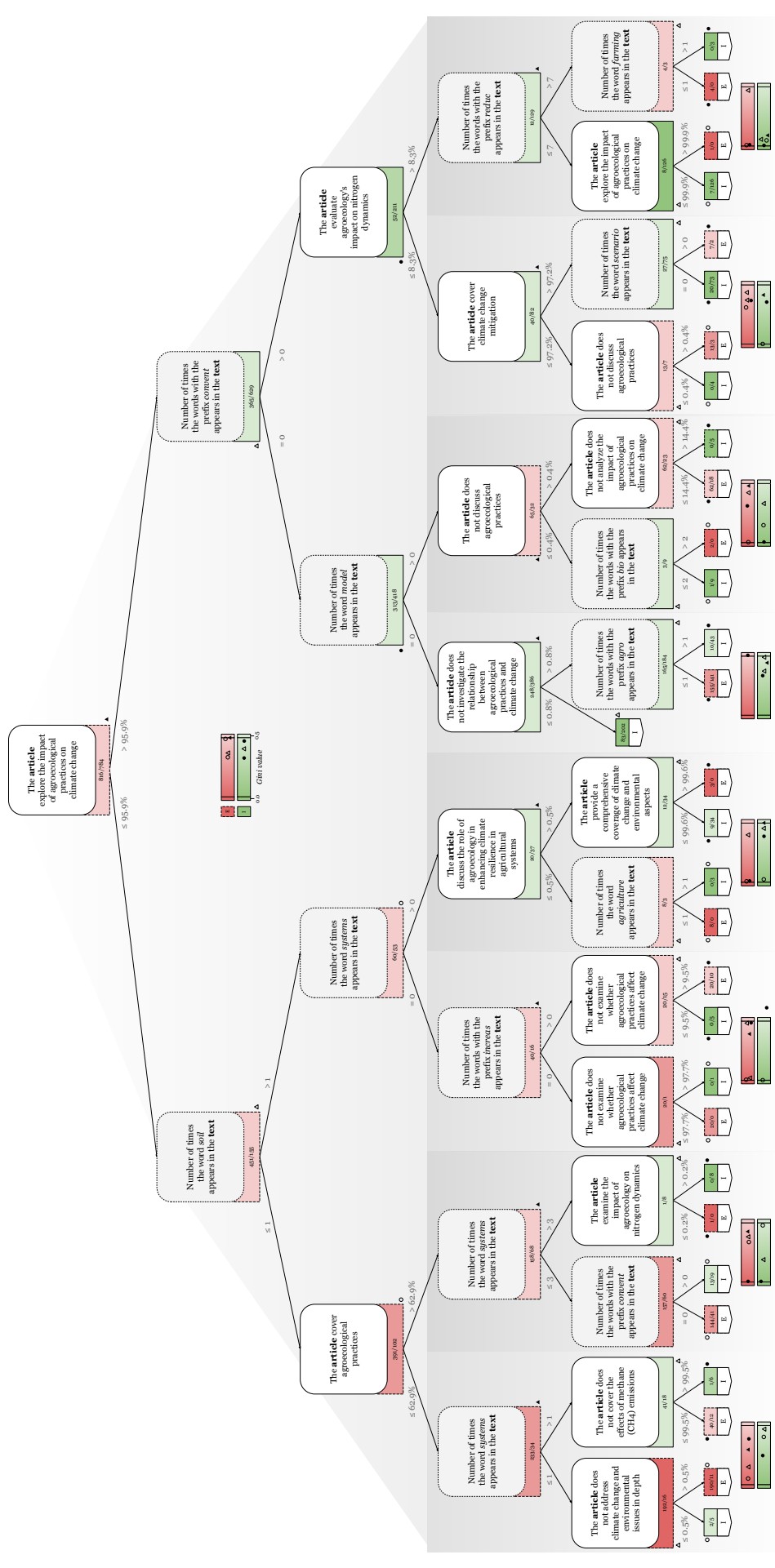

Figure 16: Decision tree with NLLF and LF for the SAC task. I is for Include, and E for Exclude. The NLLF are in white and the EF are in grey. The numbers down the boxes are the threshold of the decision. The Gini value of a tree node corresponds to the ratio of the minoritarian class elements over the dominant ones in the subtree.