# OpenReview forum: "Deep Natural Language Feature Learning for Interpretable Prediction"
_EMNLP/2023/Conference — EMNLP 2023 Main_

### Official Review · Reviewer_kRx7 · 2023-08-04

**Typos Grammar Style And Presentation Improvements:** N/A
**Soundness:** 3

**Excitement:**

4: Strong: This paper deepens the understanding of some phenomenon or lowers the barriers to an existing research direction.

**Missing References:**

N/A

**Paper Topic And Main Contributions:**

The authors proposed a method of breaking down a main task into a set of intermediary sub-tasks which are formulated as binary questions that are related to the final target task. Accordingly, the authors provide a way of understanding the decision-making process that occurs in deep learning models.


**Questions For The Authors:**

1) Is there any chance of using filtering mechanisms in the weakly labeled data (BSQs with LLMs labeled data) to improve the performance further? Is it okay to use all LLMs labeled data on BSQs to train BERT? This is because it is possible that LLMs generate noisy labeled data which degrade the performance if directly used.


**Reasons To Accept:**

The idea is very novel. The idea is very interesting in terms of decomposing tasks into sub-tasks.


**Reasons To Reject:**

The proposed method is somewhat complicated to apply to real-world scenarios. While it presents innovative concepts and potential benefits, its implementation may pose challenges for practical use. The intricate nature of the approach demands a high level of expertise and resources, which could limit its adoption in real-life applications.

**Reproducibility:**

3: Could reproduce the results with some difficulty. The settings of parameters are underspecified or subjectively determined; the training/evaluation data are not widely available.

**Reviewer Confidence:**

3: Pretty sure, but there's a chance I missed something. Although I have a good feel for this area in general, I did not carefully check the paper's details, e.g., the math, experimental design, or novelty.

---

> ### Author Rebuttal · Authors · 2023-08-28
>
> We thank the reviewer for her/his relevant comments, which we address in the following replies.
>
> **About the complexity to apply to real-world scenarios**
>
> We really want our method to be used in practice by users like in our case it could be math teachers or agronomy researchers. The implementation of our method relies on well-known python libraries like `scikit-learn`, huggingface's `transformers` and the OpenAI python API. Our framework is actually easily applicable to any new dataset, using our code made of adaptable python objects for programmers or one-liners for non-expert. We will release it after the review phase, here is a first anonymous version: https://anonymous.4open.science/r/nllf-emnlp-2023-E07C/
>
> **Question 1**:
>
> That's a very good proposition. We show in Table 3 that indeed the weak labels are not perfect, as we compared them with ones made by a human expert (78% accuracy for chatGPT that produces them). The weak labels obtained from the LLM are better than the ones generated by a BERT trained on them (68% of accuracy).
>
> For example, a method like CleanLab [1] could be used to detect the noisy labels within the weak labels and filter or correct them. We will add this to the future work section. If reviewer kRx7 wish so we can
>
> We could also use CoT in order to create better weak labels with ChatGPT. In fact, we tried this on the 100 examples from Table 3 and obtained the following results (Accuracy first, then Precision, Recall, F1):
>
> ChatGPT (already in paper): 78
>
> Yes: 71 89 79
>
> No: 88 68 77
>
> CoT: 79
>
> Yes: 78    85    81
>
> No: 81    72    76
>
> It allows to slightly improve the results, but as we tested over 100 manually labeled examples only this is not significant. Using OpenIA comes with a cost, as the prompts and the outputs are longer (in average 0.104 USD more per thousand queries).
>
> Finally, we show it is okay to use all LLMs labeled data on BSQs to train BERT. Even though our future work will focus on increasing the quality of weak labels, our method still allows reaching competitive results. Other interesting details about the weak labels are discussed in the paper in the subsection “Validation by an Expert”. First, we show the BERT actually compensates for errors of the ChatGPT. Second, the decision tree also can compensate for errors of the BERT, because it takes as input the BERT’s last layer before the softmax.
>
>
> [1] C. G. Northcutt, L. Jiang, and I. L. Chuang, “Confident learning: Estimating uncertainty in dataset labels,” J. Artif. Intell. Res., vol. 70, pp. 1373–1411, 2021.

---

### Official Review · Reviewer_wign · 2023-08-07

**Soundness:** 4

**Excitement:**

4: Strong: This paper deepens the understanding of some phenomenon or lowers the barriers to an existing research direction.

**Missing References:**

Missing the line of work in the interpretable models and the discussion of the difference between explainability and interpretability.

**Paper Topic And Main Contributions:**

The authors introduce a novel framework for interpretable classification. The framework first leverages LLM to extract Binary Subtask Questions(BSQs) (the attributes) and then fine-tunes a smaller BERT-like model with the text and BSQs to get Natural Language Learned Feature (NLLF). In addition to man-crafted and linguistic features, the authors propose to use the NLLF to improve the performance of the decision tree. The results show that the proposed framework could have better performance than black box BERT-like models and ChatGPT with/without CoT and at the same time the proposed model has fewer parameters and better explainability.

The main contributions are: (i) the proposed method is interpretable; (ii) the computation cost is low; (iii) the performance outperforms the sota models on 2 binary classification tasks.

**Questions For The Authors:**

A: Authors mention that the proposed method can be applied to any predictive task using text as input. Have you tried other more complex tasks other than binary classification? Have you tried more high-stake problems like multi-hop reasoning? If yes, how is the result comparable with sota?

B: Have you tried other explainable models? How is the result compared with yours?

C: Line532-538 claims that the proposed method fosters causality by allowing the user to write the features directly as the input, and thus this method could prevent the model from relying too heavily on latent correlational patterns. I don't agree with this statement. Feature selection and tree construction still rely on data distribution. You cannot guarantee the usage of these man-crated features. It is still possible that some high-correlation features, such as length, are artifacts.

D: ChatGPT version: Line 408 mentioned the used version is 03/23/23 ChatGPT (without the model name), while the citation refers to GPT-4. Which specific version of ChatGPT is used in the experiments?

**Reasons To Accept:**

It is an interesting work towards interpretable prediction. The authors propose to utilize LLM to extract attributes and use the data generated by LLM to train BERT-like model to obtain Natural Language Learned Feature (NLLF). NLLF and other man-crafted features are used in the decision tree which has more explainability than the black-box models. The experimental results are promising which outperform the sota models (ChatGPT with/without CoT and BERT-like model).

**Reasons To Reject:**

- The authors only conduct experiments on 2 binary datasets: Scientific Abstract Classification (SAC) and Incoherent Answer Detection (IAD). These two tasks are simple and, in my view, not high-stake enough. It is still doubt that how the framework works on other difficult tasks.
- The paper introduces explainable AI in the introduction; however, the topic of the paper is interpretable methods. The authors do not explain the definition of interpretability and do not explain the difference between these two concepts. The authors also do not provide the work in this line.
- In the experiment, the only comparison model with explainability is ChatGPT with CoT. It is better to see at least one more explainable model. Also, self-ask* performs better in high-stake questions. It would be better if there are some comparisons with self-ask.

*: Press, Ofir, et al. "Measuring and narrowing the compositionality gap in language models." arXiv preprint arXiv:2210.03350 (2022).

**Reproducibility:**

3: Could reproduce the results with some difficulty. The settings of parameters are underspecified or subjectively determined; the training/evaluation data are not widely available.

**Reviewer Confidence:**

3: Pretty sure, but there's a chance I missed something. Although I have a good feel for this area in general, I did not carefully check the paper's details, e.g., the math, experimental design, or novelty.

**Typos Grammar Style And Presentation Improvements:**

- The model in Table 1 is better to be BERT instead of BETO to be consistent with the main content (line349 and line432). BETO is only for IAD dataset, maybe it is better to use BETO/BERT.

- Line 282 The first word should be capitalized.

- It would be better to keep the same order when referring to the two datasets. Sometimes authors mentioned SAC before IAD (line305-336, line366-375, line392-401, etc), but sometimes vice versa (line161-168, Table1, line511~521, etc).

- Line 470: there should be a space in ‘IADtask’

- Table2: task should be IAD

- 4.3 is in the main content; however the visualization figures (Figure 11-14) are all in the appendix. It is better to keep them in the same page, e.g. provide details in the appendix and provide simple descriptions in the main content.

- Line535: prevents the model from relying.....

- Line899: ‘we’ instead of ‘We’

---

> ### Author Rebuttal · Authors · 2023-08-28
>
> We thank the reviewer for reading our work carefully and for her/his useful and relevant comments to improve it.
>
> ###  About the reasons to reject
>
> **Interpretability vs explainability**
>
> Indeed, we had not addressed the distinction between interpretability and explainability, and we have now added this in the CR version. We based our definitions on the _Nature Machine Intelligence_ paper of Rudin (2019), where the author makes the difference between explaining a black box model with interpreting a white-box model, or interpreting a model that outputs predictions as par with explanations.
>
> Explainability relies on algorithms aiming to explain the model predictions by showing cues to the user like LIME, or other ad-hoc methods (Fel et al., 2022; Colin et al., 2022). Interpretability relies on the possibility to know exactly why the model is making a prediction because they are inherent to the prediction and faithful to what the model actually computes.
>
> However, methods like CoT which should be interpretable (because outputting explanations with their predictions) have not always shown to give faithful explanations (Radhakrishnan, 2023). It is also arguable that our method relies on learned representations from a BERT, which decreases its overall interpretability.
>
> We will add more details (including the ones answered above) in the introduction and related works in order to make our point more clear. We also modify Table1 to change Explainability by Interpretability.
>
> Cynthia Rudin. 2019. Stop explaining black box machine learning models for high stakes decisions and use interpretable models instead. Nature machine intelligence, 1(5):206–215.
>
> A. Radhakrishnan et al., “Question Decomposition Improves the Faithfulness of Model-Generated Reasoning,” 2023.
>
> T. Fel et al., “CRAFT: Concept Recursive Activation FacTorization for Explainability,” in CVPR, 2023.
>
> J. Colin, T. Fel, R. Cadene, and T. Serre, “What I Cannot Predict, I Do Not Understand: A Human-Centered Evaluation Framework for Explainability Methods,” Adv. Neural Inf. Process. Syst., no. April, 2022.
>
> F. Jourdan, A. Picard, T. Fel, L. Risser, J. M. Loubes, and N. Asher, “COCKATIEL: COntinuous Concept ranKed ATtribution with Interpretable ELements for explaining neural net classifiers on NLP tasks,” 2023.
>
>
> **Other baselines**
>
> We thank you for the useful reference. In the CR version, we will add comparison with self-ask [1] for both our datasets and using GPT3.5-turbo as a backbone. We will also add the templates that we used in the appendix as we did for all the other configurations. The results are the following (Prec., Rec. and F1 like in Table 1):
>
> IAD:
>
> 0 shot: 21.29, 82.35, 33.84
>
> 4 shots: 51.71, 77.94, 62.17
>
> SAC:
>
> 0 shot: 63.65, 55.15, 48.13
>
> 4 shots:  70.31, 62.42, 59.50
>
> This makes self-ask a better baseline than Chain of Thoughts for IAD (62.2 vs 56.5), but not for SAC (59.5 vs 62.7). We can implement  and add other recent baselines for the CR version, like self-consistent chain of thoughts [2] or tree of thoughts [3] if the reviewers wish so.
>
> [1] O. Press, M. Zhang, S. Min, L. Schmidt, N. A. Smith, and M. Lewis, “Measuring and Narrowing the Compositionality Gap in Language Models,” 2022.
>
> [2] X. Wang et al., “Self-Consistency Improves Chain of Thought Reasoning in Language Models,” pp. 1–24, 2022.
>
> [3] S. Yao, D. Yu, J. Zhao, and T. L. Griffiths, “Tree of Thoughts : Deliberate Problem Solving with Large Language Models,” no. 1, pp. 1–11.
>
> **Task difficulty**
>
> Our method has been applied here in the context of binary classification tasks, but it can be applied to any multinomial classification or regression. However, it is important to mention that the tasks we tackled in our work are not that easy: for SAC a BERT model is far from reaching high performances like 95% of Accuracy (like in SST). Moreover we show that these tasks are overly difficult for a LLM like GPT-3.5turbo using ICL.
>
> ###  Questions to authors
>
> **Answer A**: We stated in l145-147: “Indeed, it can be applied to any problem that can be reasonably decomposed in simpler tasks”. This was a mistake on our part: it can be applied to any classification or regression problem, but the proposed method cannot be applied as it is now to a generative task! We will rephrase this.
>
> We did not try any other tasks than binary classification, but our method is indeed applicable to any multinomial classification or even regression. We tried our setting on these two binary classification tasks where the practitioners (teachers and agronomy researchers) would like to have more explanations on the prediction of the machine learning models, and we believe this applied setting is interesting.
> We haven't tried our method on multi-hop reasoning yet as this was not the focus of our work (and we find our paper already very dense and complex for 8 pages). However, we are currently working on extensions of our work, and multi-hop reasoning is a good direction for future work, with many adaptations though. We will add this information to the future work section.
>
> **Answer B**: We conducted new experiments with the self-ask method, as stated above. We can add other baselines for the CR version if you wish so.
>
> Finally, we are also right now running experiments with other explainable post-hoc methods for classifiers like LIME, SHAP and COCKATIEL (Jourdan et al., 2023), asking humans their preferences between the different type of explanations, based on questionnaires similar to (Jesus et al., 2021). However we believe our paper is dense enough to keep this for a future publication as this drives many results and analyses over the results and the models. We add this in the future work section.
>
> F. Jourdan, A. Picard, T. Fel, L. Risser, J. M. Loubes, and N. Asher, “COCKATIEL: COntinuous Concept ranKed ATtribution with Interpretable ELements for explaining neural net classifiers on NLP tasks,” 2023.
>
> S. Jesus et al., “How can i choose an explainer?: An Application-grounded Evaluation of Post-hoc Explanations,” FAccT 2021 - Proc. 2021 ACM Conf. Fairness, Accountability, Transpar., pp. 805–815, 2021.
>
> **Answer C**: This is a very good point, we agree with the reviewer that feature selection and tree construction still rely on data distribution and are thus not completely causal. We will rephrase these lines to reflect this, and remove the initial claim.
>
> One way of testing our hypothesis would be to construct a synthetic dataset that allows us to test it, for example in a dataset where the bias has been deliberately inflated (Reif et al., 2023). Due to time constraints, we leave this task for future development and assessment of our model. Nevertheless we are adding details in our paper regarding this issue
>
> Y. Reif and R. Schwartz, “Fighting Bias with Bias: Promoting Model Robustness by Amplifying Dataset Biases,” in Findings of ACL: ACL 2023, 2023, pp. 13169–13189.
>
> **Answer D**: We used ChatGPT (GTP3.5-turbo, version from 23 March 2023) and not GPT4, this was a mistake, thanks for spotting it, we changed the citation.
>
> ###  Typos
> Finally, we took into account all of your comments regarding the typos and presentation. Thanks.

---

### Official Review · Reviewer_rrHR · 2023-08-09

**Soundness:** 4

**Excitement:**

3: Ambivalent: It has merits (e.g., it reports state-of-the-art results, the idea is nice), but there are key weaknesses (e.g., it describes incremental work), and it can significantly benefit from another round of revision. However, I won't object to accepting it if my co-reviewers champion it.

**Paper Topic And Main Contributions:**

The authors have proposed a novel technique for breaking down the main task into intermediary sub-tasks, which is quite impressive.

**Questions For The Authors:**

1) Why well-known baselines are not implemented?

2) Why the limitations section doesn't contain above time complexity and regarding cost-effective solution. I think the proposed algorithm is not a cost effective solution neither faster at training and generating predictions.

3) If the dataset is large. Will the proposed algorithm work?

**Reasons To Accept:**

+ The idea of breaking the main task into intermediatory sub tasks is quite interesting and can be helpful for the further research.

+ The visualizations and diagrams are quite expressive.

**Reasons To Reject:**

- The experiments conducted didn't involve well-known baselines.

- The proposed algorithm also takes time and needs computational resources for huge datasets. This point should be mentioned in the limitations section.

**Reproducibility:**

4: Could mostly reproduce the results, but there may be some variation because of sample variance or minor variations in their interpretation of the protocol or method.

**Reviewer Confidence:**

4: Quite sure. I tried to check the important points carefully. It's unlikely, though conceivable, that I missed something that should affect my ratings.

---

> ### Author Rebuttal · Authors · 2023-08-28
>
> We would like to thank reviewer rrHR for the feedback, we will try to answer all of your questions and comments.
>
> **Regarding the time and complexity costs**
>
> Our method relies on three components: an LLM – GPT3.5-turbo, a medium-size LM – BERT, and a very small model – decision tree. Regarding the cost of the method:
> * Only the decision tree is trained over the full dataset. This is cost effective and very quick (the max depth of the tree is set to 5)
> * The medium-size LM is trained only over $p_l$ = 10% of the dataset (per $C=10$ BSQ). This has the same cost as training a BERT (we did not even search for the best hyperparameters). Because we add the Binary Subtask Question in the input string, it adds some 10-15 tokens (which we think is negligible).
> * The LLM is used to weakly label $p_l$ = 10% of the dataset (per $C=10$ BSQ) in a zero-shot way.The output of the LLM is in the form of a yes/no answer, hence it is the costless setting when using chatGPT. The price depends on the number of generated tokens, and of the input tokens: it is far cheaper to do this than using 4-shot CoT for example.
>
> The initial part to create the Binary Subtask Question used only pq percent (0.015% for IAD, 1.688% for SAC) of the data, which is very small, 27 examples in both cases.
>
> We will add in the limitation section that it needs computational resources for huge datasets because our method relies on a LLM like chatGPT (GPT3.5-turbo), like any of the methods relying on LLM or LLM themselves. Nevertheless, we know that in the future, smaller open-source alternatives like Alpaca, Vicuña or PaLM-2 will drastically reduce this cost.
>
> We will also add in the CR version the cost in dollars of each experiment, which in total is less than two dollars for both experiments: 1.264 USD for SAC and 0.188 USD for AID.
>
> At inference time, the already trained BERT needs to generate only the NLLF that were selected (14 for SAC, 13 for IAD). This is 14 times more costly than a BERT, but far less than an LLM. Extracting the handcrafted features and running the decision tree is very fast. We will surely add these details that we forgot to mention in the Limitation section.
>
> **About the other baselines**
>
> Following your remarks and the ones of reviewer wign, we added comparison with other baselines, and not only CoT. We compared our system with self-ask [1] (using GPT3.5-turbo as a backbone). The results are the following (Prec., Rec. and F1 like in Table 1):
>
> IAD:
>
> 0 shot: 21.29, 82.35, 33.84
>
> 4 shots: 51.71, 77.94, 62.17
>
> SAC:
>
> 0 shot: 63.65, 55.15, 48.13
>
> 4 shots:  70.31, 62.42, 59.50
>
> This makes self-ask a better baseline than Chain of Thoughts for IAD (62.2 vs 56.5), but not for SAC (59.5 vs 62.7). We can implement  and add other recent baselines for the CR version, like self-consistent chain of thoughts [2] or tree of thoughts [3] if the reviewers wish so.
>
> Finally, we believe that the BERT baseline is strong enough as a baseline for text classification. Can you be more precise in what you were expecting so we can answer this point more precisely and add other baselines that you would like to see?
>
> [1] O. Press, M. Zhang, S. Min, L. Schmidt, N. A. Smith, and M. Lewis, “Measuring and Narrowing the Compositionality Gap in Language Models,” 2022.
>
> [2] X. Wang et al., “Self-Consistency Improves Chain of Thought Reasoning in Language Models,” pp. 1–24, 2022.
>
> [3] S. Yao, D. Yu, J. Zhao, and T. L. Griffiths, “Tree of Thoughts : Deliberate Problem Solving with Large Language Models,” no. 1, pp. 1–11.
>
> **About the dataset size**
>
> Finally, there is no reason why our algorithm should not work on datasets bigger than 15k training examples (IAD). With more data, it is possible to get more BSQs, improve the performances of the NLLF Generator, and eventually augment the depth of the decision tree.

---

### Meta-Review · Area_Chair_XviM · 2023-09-18

**Recommendation:** 4

**Metareview:**

This paper presents a general strategy for decomposing a complicated task into a set of easier sub-tasks. Specifically, the proposed method represents each example with a vector consisting of a list of binary questions. By this decomposition, the proposed method not only improves prediction performance but also can generate more interpretable predictions (or be used as inputs to interpretable machine learning models). The reviewers commonly agreed that the proposed method is interesting and novel, although expressed their concerns about the experiment setup (e.g., baseline methods) as well as whether the method is generally applicable to real-world applications.

---

### Decision · Program_Chairs · 2023-10-07

**Decision:**

Accept-Main

**Comment:**

This paper presents a general strategy for decomposing a complicated task into a set of easier sub-tasks. Specifically, the proposed method represents each example with a vector consisting of a list of binary questions. By this decomposition, the proposed method not only improves prediction performance but also can generate more interpretable predictions (or be used as inputs to interpretable machine learning models). The reviewers commonly agreed that the proposed method is interesting and novel, although expressed their concerns about the experiment setup (e.g., baseline methods) as well as whether the method is generally applicable to real-world applications.